



# Heterogenous CO₂ and CH₄ content of glacial meltwater of the Greenland Ice Sheet and implications for subglacial carbon processes

Andrea J. Pain[1,2], Jonathan B. Martin[1], Ellen E. Martin[1], Shaily Rahman[1,3]

[1] University of Florida, Department of Geological Sciences, Gainesville, FL 32611
[2] Now at: University of Maryland Center for Environmental Science, Horn Point Lab, Cambridge, MD 21613
[3] Now at: University of Southern Mississippi, Department of Marine Science, Stennis Space Center, MS 39529

*Correspondence to*: Andrea J. Pain (ajpain@ufl.edu)

**Abstract.** Accelerated melting of the Greenland Ice Sheet (GrIS) has increased freshwater delivery to the Arctic Ocean and
amplified the need to understand the impact of GrIS meltwater on Arctic greenhouse gas (GHG) budgets. We measured carbon
dioxide ($CO_2$) and methane ($CH_4$) concentrations and $\delta^{13}C$ values and use geochemical models to evaluate subglacial $CH_4$ and
$CO_2$ sources and sinks in water discharging from three subglacial outlets of the GrIS in southwest (Isunnguata and Russell
Glaciers) and southern Greenland (Kiattut Sermiat). $CH_4$ concentrations vary by orders of magnitude between sites and are
saturated with respect to atmospheric concentrations at Kiattut Sermiat, but are supersaturated at southwest sites, even though
oxidation reduces concentrations by up to 50% during periods of low discharge. $CO_2$ concentrations range from supersaturated
at Isunnguata to undersaturated at Kiattut Sermiat. $CO_2$ is consumed by mineral weathering throughout the melt season at all
sites, however differences in the magnitude of subglacial $CO_2$ sources result in meltwaters that are either sources or sinks of
atmospheric $CO_2$. The predominant source of $CO_2$ at Isunnguata is organic matter (OM) remineralization, but Russell and
Kiattut Sermiat sites have multiple or heterogeneous subglacial $CO_2$ sources that maintain atmospheric $CO_2$ concentrations at
Russell but not at Kiattut Sermiat where $CO_2$ is undersaturated. These results highlight the variability in GHG dynamics under
the GrIS. Constraining this variability will improve our understanding of the impact of GrIS melt on Arctic GHG budgets, as
well as the role of continental ice sheets in GHG variations over glacial-interglacial timescales.

## 1 Introduction

Glaciers play an important role in global chemical cycles due to the production of fine-grained sediments that
participate in carbonate and silicate mineral weathering reactions (Table 1), which are the principal sink of atmospheric $CO_2$
over geologic timescales (Berner et al., 1983; Walker et al., 1981). Variations in the weathering intensity of comminuted
sediments may contribute to glacial-interglacial atmospheric $CO_2$ variations as sediments are alternately covered by ice and
exposed following ice retreat. However, the importance of $CO_2$ consumption by mineral weathering is poorly understood,
including effects from the advance and retreat of continental ice sheets (Ludwig et al., 1999). Recent evaluations of carbon
budgets in proglacial environments indicate that mineral weathering results in net sequestration of atmospheric $CO_2$, suggesting



that proglacial systems are underrecognized as Arctic $CO_2$ sinks (St. Pierre et al., 2019), however alternate processes could lead to the production of greenhouse gases (GHG) in glacial systems. For instance, $CH_4$ production in anaerobic subglacial environments driven by the remineralization of organic matter (OM) contained in soils and forests covered during glacial margin fluctuations has been suggested as a potential carbon feedback to drive warming (Sharp et al., 1999; Wadham et al.,

2008). Because the global warming potential of $CH_4$ is 25 times greater than $CO_2$, even limited subglacial methanogenesis has the potential to strongly impact the GHG effect of glacial meltwater. Combined inorganic and organic subglacial processes may therefore produce glacial meltwater that is a source or sink of GHG. While the net impact of these processes on modern carbon fluxes is poorly constrained, determining these impacts will improve modern carbon budgets as well as depictions of how fluxes may have evolved during the advance and retreat of continental ice sheets.

In subglacial environments where remineralization is limited by low OM availability, the major element solute load of glacial meltwater is typically dominated by products of mineral weathering reactions (Tranter, 2005). The extent of mineral weathering in subglacial environments depends in part on the availability of acids to drive reactions, namely sulfuric and carbonic acids (Table 1). Sulfuric acid is derived from the oxidation of reduced sulfur species, which largely occur as iron-sulfide minerals including pyrite (Tranter, 2005). Sulfide oxidation may occur abiotically, however the kinetics of microbially

mediated sulfide oxidation is several orders of magnitude faster and may lead to local depletion of oxygen provided sufficient supply of sulfide minerals (Sharp et al., 1999). In contrast, carbonic acid may be derived from multiple external or *in situ* sources of $CO_2$ to the system. The dominant external source is supraglacial meltwater that flows to the subglacial system through moulins following equilibration with atmospheric $CO_2$ (Fig. 1). Unlike proglacial environments where free exchange between water and the atmosphere may resupply $CO_2$ consumed by weathering, subglacial environments may be partially or

fully isolated from the atmosphere, limiting further atmospheric $CO_2$ invasion and thus the extent of mineral weathering with carbonic acid. However, additional atmospheric $CO_2$ may be delivered in open portions of the subglacial environment though exchange in fractures or moulins along subglacial flow paths or in partially air-filled conduits, allowing a much greater magnitude of carbonic acid mineral weathering (Graly et al., 2017). $CO_2$ may also be derived from *in situ* sources, such as gaseous $CO_2$ contained in ice bubbles of basal ice, or fluid inclusions in rocks that release volatiles (including $CO_2$) following

mechanical grinding (Macdonald et al., 2018). When OM is available, its remineralization also generates $CO_2$ (and potentially $CH_4$) along with nutrients, but low OM availability in many subglacial systems limits remineralization as a $CO_2$ source (Fig. 1).

         The role of subglacial carbon processes may play an increasingly important role in modern Arctic carbon budgets as disproportionate warming increases glacial meltwater and sediment fluxes to the ocean, particularly from the Greenland Ice

Sheet (GrIS). The Greenland Ice Sheet (GrIS) is the last remaining ice sheet in the Northern hemisphere following collapse of all others since the Last Glacial Maximum (~25 ka). It has been losing mass at increasing rates that averaged 286±20 Gt/yr between 2010-2018, representing a six-fold increase since the 1980s (Mouginot et al., 2019). While mineral weathering significantly modifies the composition of GrIS subglacial discharge (e.g. Hindshaw et al., 2014; Deuerling et al., 2018; Urra



et al., 2019) and should consume $CO_2$ similar to other glacial and proglacial environments, the recent identification of microbially driven reactions (including methanogenesis) in subglacial environments of the GrIS indicates that organic processes may also play a role (Dieser et al., 2014; Lamarche-Gagnon et al., 2019; Musilova et al., 2017). The relative importance of subglacial GHG sinks ($CO_2$ consumption through mineral weathering) and sources (such as OM remineralization) determine the GHG composition of subglacial discharge, which may then serve as a source or a sink of atmospheric GHGs. Constraining the relative impacts and variability of these processes underneath the GrIS will provide important information regarding the current and future impact of GrIS loss on Arctic carbon budgets, as well the role of continental ice sheets on carbon cycle feedbacks.

To evaluate the net impact of carbon processes on the GHG composition of subglacial discharge of the GrIS, we compare water chemistry, dissolved $CO_2$ and $CH_4$ concentrations, and gas stable isotopic compositions between three subglacial discharge sites draining land-terminating glaciers of the GrIS over the melt seasons of 2017 and 2018 (Fig. 2). We employ mass balance models utilizing the concentrations of major cations and anions to determine the magnitude of the impact on $CO_2$ concentrations from mineral weathering reactions (Table 1). These results are combined with measured gas concentrations to determine the relative importance of mineral weathering compared to OM remineralization on the $CH_4$ and $CO_2$ content of subglacial discharge. We also assess the temporal and spatial variability of these processes under the GrIS to improve our understanding of carbon cycling in Greenland subglacial environments and the implications of GrIS mass loss on Arctic carbon budgets.

## 2 Methods

### 2.1 Study locations

Our three subglacial discharge locations are located in southwest (Fig. 2a, b) and southern (Fig. 2a, c) Greenland. The Isunnguata Glacier (IS; 67°09'27.1" N, 50°03'25.0" W) and Russell Glacier (RU; 67°05'22.1" N, 50°14'18.7" W) drain to the Watson River, which is one of the largest proglacial rivers in Greenland. Watson River discharge is monitored by PROMICE (Programme for Monitoring of the Greenland Ice Sheet; van As et al., 2018) and total discharge was 4.3 and 3.6 km³ of water in 2017 and 2018, respectively (van As et al., 2018). The total catchment size for the Isunnguata is 15,900 km², though our water samples were collected from a smaller sub-catchment with a drainage area of approximately 40 km² (Lindbäck et al., 2015; Rennermalm et al., 2013). The total drainage area for the Russell glacier is estimated at 300 km². This estimate comes from subtracting the Leverett drainage area estimated at approximately 600 km² (Hawkings et al., 2016) from the approximately 900 km² catchment that includes the Russell and Leverett drainages (Lindback et al., 2015). While discharge from the third site in southern Greenland, Kiattut Sermiat (KS; 61°12'13.5" N, 45°19'49.1"W), is not monitored, a previous study using dye tracing techniques estimated approximately 0.22 km³ of discharge in 2013, and its catchment size was estimated at 36 km² (Hawkings et al., 2016). Underlying lithologies differ between sites. Watson River sites are located near the boundary between the Archean Craton to the south and the southern Nagssugtoqidian Orogen to the north (Henriksen et



al., 2009). The Archean block is composed of granites and granulite facies orthogneisses that were intruded by mafic dykes during Paleoproterozoic rifting. These rocks were deformed and modified during subsequent continent-to-continent collision in the Paleoproterozoic to create the amphibolite facies gneisses of the southern Nagssugtoqidian Orogen (van Gool et al., 2002). Kiattut Sermiat lies within the Paleoproterozoic Ketilidian fold belt (Henriksen et al., 2009). Lithologies in this region

include the Julianehåb Granite and associated basic intrusions and the sedimentary and volcanic rocks of the Mesoproterozic Gardar Province that include a suite of alkaline igneous rocks and basaltic dykes with interbedded sandstones (Kalsbeek and Taylor, 1985; Upton et al., 2003).

Previous studies have characterized chemical weathering reactions in subglacial discharge to the Watson River (Deuerling et al., 2018; Hasholt et al., 2018; Yde et al., 2014), Kiattut Sermiat (Hawkings et al., 2016), and comparatively

between these sites (Urra et al., 2019). There has been extensive work regarding ice sheet dynamics and hydrology in the Watson River catchment (Van As et al., 2017, 2018; Lindbäck et al., 2015) as well as Kiattut Sermiat (Warren and Glasser, 1992; Winsor et al., 2014). Previous studies near these study locations have identified $CH_4$ and $CO_2$ supersaturation in subglacial discharge of the Isunnguata site (Christiansen and Jørgensen, 2018; Ryu and Jacobson, 2012), while methanogenic microbial communities have been observed at Russell Glacier (Dieser et al., 2014) and $CH_4$ supersaturation at the Leverett

Glacier (Lamarche-Gagnon et al., 2019), which also flows into the Watson River (Fig. 2c). Subglacial permafrost has been identified near the Isunnguata site (Ruskeeniemi et al., 2018) and attributed to Holocene fluctuations in the ice sheet margin. While a similar Holocene ice retreat and re-advance may have occurred in southern Greenland (Larsen et al., 2016), it is unknown whether this retreat led to the formation of organic deposits.

**2.2 Sample collection**

We collected water samples from subglacial discharge sites in spring and fall of 2017, and the summer of 2018 to observe seasonal variations in water composition. Samples were collected as close as possible to the subglacial discharge site, which was less than 10 m for the Isunnguata site, approximately 100 m for the Russell glacier site, and approximately 1.1 km for the Kiattut Sermiat site (Fig. 2). We collected water samples by pumping water through a 0.5-cm flexible PVC tube that was placed in flowing water as far as possible from shore (approximately 1-2 m). A YSI Pro-Plus sensor that was calibrated

daily was installed in an overflow cup filled from the bottom to measure specific conductivity (Sp.C), temperature, pH, dissolved oxygen, and oxidation-reduction potential (ORP). These parameters were monitored until stable, between about 10 and 30 minutes, after which samples were collected and preserved in the field according to the solute to be measured after being filtered through a 0.45 µm trace-metal grade Geotech high capacity disposable canister filter. Samples for cations and anions were collected in HDPE bottles; cation samples were preserved with Optima-grade ultrapure nitric acid (pH<2) while

no preservative was added to anion samples. Samples for ammonium ($NH_4$) were filtered into 15 mL polypropylene containers and frozen until analysis. Dissolved inorganic carbon (DIC) samples were filtered through 0.2 µm filters directly to the bottom of 20 ml Qorpac glass vials and allowed to overflow until sealed tightly with no headspace.



Gas samples were collected in duplicate via headspace extractions according to methods outlined in Repo et al. (2007) and Pain et al. (2019). Unfiltered water was pumped into the bottom of 500 mL bottles until they overflowed. Bottles were

immediately capped with rubber stoppers fitted with two 3-way inlet valves. 60 mL of water was extracted from one inlet and replaced with 60 mL of atmospheric air (for spring and fall 2017 sampling trips) or ultrapure $N_2$ gas in a gas bag (summer 2018 sampling trip). Bottles were shaken for 2 minutes to equilibrate headspace gas with water, and headspace gas was extracted and immediately injected into 60 ml glass serum bottles that had been evacuated immediately prior to sample introduction. Samples were stored at room temperature until analysis, which occurred within one week of collection. Measured

headspace concentrations were converted to dissolved concentrations using methods outlined in Pain et al. (2019). When atmospheric air was used for headspace extractions, atmosphere samples were collected in tandem and analyzed to correct each sample for calculated dissolved $CO_2$ and $CH_4$ concentrations and isotopic compositions. This correction altered $CH_4$ concentrations by up to 22% for one sample from the Russell glacier, though less than 5% for all other samples, and resulted in a correction of $\delta^{13}C$-$CH_4$ of up to 1.3‰. For $CO_2$, the correction altered concentrations by up to 15% for one sample collected

at Kiattut Sermiat, though less than 10% for all other samples, and resulted in a correction of $\delta^{13}C$-$CO_2$ of up to 0.4‰.

For fall 2017 and summer 2018 sampling trips, alkalinity was measured in the field laboratory within 3 days of collection by titration with 0.01 N HCl using the Gran method. Because alkalinity measurements were not available for the spring 2017 sampling trip, we estimate alkalinity with PHREEQc modeling and the phreeqc.dat database (Parkhurst, 1997) using major cations and anions, pH, temperature, and DIC concentrations as model inputs.

**2.3 Laboratory analysis**

Gas samples were analyzed for $CO_2$ and $CH_4$ concentrations, and $\delta^{13}C$-$CO_2$ and $\delta^{13}C$-$CH_4$ on a Picarro G2201-i cavity ring-down spectrometer. Carbon isotopic compositions are reported in reference to Vienna Pee Dee Belemnite (VPDB). Check standards of known $CO_2$ and $CH_4$ concentrations and isotopic compositions were measured during each sample run and were accurate within 10%. Anion and cation concentrations were measured on an automated Dionex ICS-2100 and ICS-1600 Ion

Chromatograph, respectively. Error on replicate analyses was less than 5%. DIC concentrations were measured on a UIC (Coulometrics) 5011 $CO_2$ coulometer coupled with an AutoMate Preparation Device. Samples were acidified and the evolved $CO_2$ was carried through a silver nitrate scrubber to the coulometer where total C was measured. Accuracy was calculated to be ±0.1 mg/L based on measurement of check standards. Dissolved ammonium ($NH_4$) concentrations were analyzed on a Seal AutoAnalyzer III. Error on check standards was less than 10%.

**2.4 Methane modeling**

To assess $CH_4$ sources and sinks, we calculate $\varepsilon_c$, or the carbon isotopic fractionation factor between $CO_2$ and $CH_4$ as defined in Whiticar (1999):

$$\varepsilon_c = \delta^{13}C_{CO2} - \delta^{13}C_{CH4} \tag{5}$$



Values of $\varepsilon_c$ reflect methanogenesis pathways (acetoclastic or $CO_2$ reduction) as well as the extent of oxidation.
Values of $\varepsilon_c$ between approximately 40 and 55‰ are produced for $CH_4$ produced via acetoclastic methanogenesis, while $CO_2$
reduction produces values between approximately 55 and 90‰. Lower values ($\varepsilon_c$ between 5 and 30) result when $CH_4$ oxidation
predominates. Modern atmospheric input without additional alteration of $CO_2$ or $CH_4$ isotopic systematics results in a $\varepsilon_c$ value
of approximately 40 (Whiticar, 1999).

We calculated $CH_4$ oxidation using the isotopic method outlined in Mahieu et al. (2008) and Preuss et al. (2013). The
fraction of oxidized methane ($f_{ox}$) in an open system is given by:

$$f_{ox} = \frac{\delta_E - \delta_P}{1000 \cdot (\alpha_{ox} - \alpha_{trans})} \tag{6}$$

where $\delta_E$ is the measured $\delta^{13}C$-$CH_4$ value for each water sample, $\delta_P$ is $\delta^{13}C$-$CH_4$ of produced methane, $\alpha_{ox}$ is the oxidation
fractionation factor, and $\alpha_{trans}$ is a fractionation factor resulting from diffusive transportation of $CH_4$. While the exact value of
$\delta_P$ is unknown, diagenetic alteration of $\delta^{13}C$-$CH_4$ values through oxidation or transport only enrich $\delta^{13}C$-$CH_4$ signatures,
therefore the value of $\delta_P$ is taken as the most depleted $\delta^{13}C$-$CH_4$ signature assuming it is the least impacted by diagenetic
alteration. Literature-reported values for $\alpha_{ox}$ range between 1.003 and 1.049. We calculate the fraction of oxidized methane
with the largest fraction factor ($\alpha_{ox} = 1.049$; Mahieu et al., 2008), which yields the minimum amount of $CH_4$ oxidation required
to explain the observed variations in $\delta^{13}CH_4$, and thus is a conservative estimate for $CH_4$ oxidation. Literature-reported values
for $\alpha_{trans}$ vary from 1 for advection-dominated systems to 1.0178 for diffusion-dominated porous media (Visscher et al., 2004;
Mahieu et al., 2008; Preuss et al., 2013). We assume that transport is advection dominated and thus assume $\alpha_{trans} = 1$.

## 2.5 Mineral weathering and carbonate modeling

We used major cation and anion concentrations and alkalinity to partition solutes into the four mineral weathering
reactions in Table 1 after correcting solute concentrations for marine aerosol deposition using measured chloride
concentrations and standard seawater element ratios. The mass balance model followed the methods of Deuerling et al. (2019).
After apportioning solutes to mineral weathering reactions, we used the stoichiometries of reactions to calculate the impact of
each reaction on dissolved $CO_2$ concentrations (Table 1). The mineral weathering model apportions solutes to reactions in
Table 1 based on the ratios of Ca/Na and Mg/Na in silicate minerals in stream bedload samples, which were taken to be 0.54
and 0.38, respectively, for Isunnguata and Russell Glacier samples (Deuerling et al., 2019; Hindshaw et al., 2014; Wimpenny
et al., 2010, 2011) and 0.39 and 0.27, respectively, for Kiattut Sermiat samples (Da Prat and Martin, 2019). Because mineral
weathering reactions may both add and remove $CO_2$, we discuss both the net impact of mineral weathering on $CO_2$
concentrations (Net $CO_{2-MW}$), which may have a positive or negative value:

$$[\text{Net } CO_{2-MW}] = [CO_{2-CarbCA}] + [CO_{2-CarbSA}] + [CO_{2-SilCA}] \tag{7}$$

as well as the total impact of mineral weathering on $CO_2$ concentrations (Total $CO_{2-MW}$),



$$[\text{Total CO}_{2\text{-MW}}] = |[\text{CO}_{2\text{-CarbCA}}]| + |[\text{CO}_{2\text{-CarbSA}}]| + |[\text{CO}_{2\text{-SilCA}}]| \tag{8}$$

where changes in the concentrations of $CO_2$ are defined by their absolute values. To discuss the relative importance of individual reactions, we define proportional contributions of each reaction as follows:

$$\%\text{CO}_{2\text{-CarbCA}} = \frac{|[\text{CO}_{2-CarbCA}]|}{[\text{Total CO}_{2-MW}]} *100 \tag{9a}$$

$$\%\text{CO}_{2\text{-CarbSA}} = \frac{|[\text{CO}_{2-CarbSA}]|}{[\text{Total CO}_{2-MW}]} *100 \tag{9b}$$

$$\%\text{CO}_{2\text{-SilCA}} = \frac{|[\text{CO}_{2-SilCA}]|}{[\text{Total CO}_{2-MW}]} *100 \tag{9c}$$

We combine measured $CO_2$ concentrations with Net $CO_{2\text{-MW}}$ in order to determine the magnitude of $CO_2$ production or consumption in the subglacial environment due to processes besides mineral weathering. This analysis assumes that the concentration of $CO_2$ measured at the subglacial outlet is equivalent to the net change in $CO_2$ due to mineral weathering plus the sum of all other subglacial $CO_2$ sources and sinks. We refer to the sum of all other subglacial $CO_2$ sources and sinks as $CO_{2\text{-total}}$, which represents the amount of $CO_2$ that must have been supplied to the subglacial environment to balance the mineral weathering $CO_2$ sink:

$$\text{CO}_{2\text{-measured}} = \text{Net CO}_{2\text{-MW}} + \text{CO}_{2\text{-total}} \tag{10}$$

The sources of $CO_2$ to $CO_{2\text{-total}}$ may be evaluated through the use of Keeling plots, which are constructed as the inverse of $CO_2$ concentrations ($[\text{CO}_2]^{-1}$) versus stable isotopic composition ($\delta^{13}\text{C-CO}_2$). If variations in the concentration and isotopic composition of $CO_2$ arise from the mixing of two $CO_2$ reservoirs with constant isotopic compositions and concentrations (Keeling, 1958), a linear relationship is expected between $[\text{CO}_2]^{-1}$ and $\delta^{13}\text{C-CO}_2$. The y-intercept of a regression between these variables represents the isotopic composition of the high-$CO_2$ end member. Because measured $CO_2$ concentrations include both subglacial $CO_2$ sources and sinks, which may include considerable consumption through mineral weathering reactions, the magnitude of the total subglacial $CO_2$ source is taken as $CO_{2\text{-total}}$. We therefore construct Keeling plots between $[\text{CO}_{2\text{-total}}]^{-1}$ and measured $\delta^{13}\text{C-CO}_2$ values because while mineral weathering impacts the concentration of $CO_2$, its isotopic composition is not appreciably altered (Myrttinen et al., 2012) compared to the range of isotopic compositions of potential $CO_2$ end members, namely OM remineralization, atmospheric $CO_2$, and lithogenic $CO_2$ sources due to mechanical grinding (Fig. 1).

### 2.6 Discharge relationships

We evaluate the relationship between subglacial $CH_4$ and $CO_2$ dynamics and discharge using discharge records provided by PROMICE (Programme for Monitoring of the Greenland Ice Sheet; van As et al., 2018). Discharge records are collected at the outlet of the Watson River, which represents the combined discharge of Isunnguata and Russell glaciers as well as other outlet glaciers including the Leverett Glacier and major tributaries including the Orkendalen River (Fig. 2).





Watson River discharge estimates are therefore greater than the true discharge of our individual sampling locations, however we assume that discharge at the Isunnguata and Russell subglacial outlet sites is roughly proportional to Watson River discharge and exhibits similar temporal variability (Rennermalm et al., 2012). Because diurnal fluctuations in river discharge can be large, and differing water travel times from subglacial outlet sites to the Watson River mouth induces a lag of up to 8
220 hours between maximum daily discharge at subglacial discharge sites and the Watson River outlet, we compare subglacial $CH_4$ and $CO_2$ concentrations to average daily discharge, calculated as the average of hourly discharge estimates over the days on which subglacial discharge water samples were collected. Because no discharge information is available for Kiattut Sermiat, we assess discharge relationships at Watson River (Isunnguata and Russell) sites only.

## 3 Results

### 3.1 Temporal variability in water chemistry and gas concentrations

Chemical parameters differ between subglacial discharge sites as well as throughout the melt season. Specific conductivity (Sp.C; Fig. 3a) is typically highest at Kiattut Sermiat (26±8 µS/cm), followed by Russell (22±5 µS/cm) and Isunnguata sites (13±9 µS/cm; Fig. 3a). All sites show variability throughout the melt season, with lowest values occurring in the summer for Isunnguata and Russell, while Sp.C drops continuously throughout the melt season for KS. Sites differ in pH,
and values at Kiattut Sermiat (8.2±0.4) are higher than both Russell (7.2±0.2) and Isunnguata (6.6±0.6; Fig. 3b), and while values vary throughout the melt season, no consistent trend is identified between sites. The saturation of dissolved oxygen (D.O.) with respect to atmospheric concentrations is similar between sites, though Isunnguata (98±8%) values fall below Russell (115±16%) and Kiattut Sermiat (117±11%) during all sampling times and exhibit undersaturation in the mid-summer, while Russell and Kiattut Sermiat are consistently supersaturated (Fig. 3c). Alkalinity is similar at Russell (93±31 µeq/L) and
Kiattut Sermiat (93±26 µeq/L) which are higher than at Isunnguata (39±25 µeq/L), but all reach minimum values in summer (Fig. 3d). $CH_4$ concentrations differ by orders of magnitude between sites (Fig. 3e) and are consistently supersaturated with respect to atmospheric concentrations at Isunnguata (648±411 ppm or 1575±997 nM) and Russell (58±33 ppm or 110±78 nM), while close to atmospheric equilibrium at Kiattut Sermiat (4±2 ppm or 9±5 nM). $\delta^{13}C$-$CH_4$ values (Fig. 3f) are similar between Isunnguata (-54.7±7.5‰), Russell (-52±7.3‰), and Kiattut Sermiat (-57.6±14.2‰). Measured $CO_2$ concentrations (Fig. 3g)
are consistently supersaturated with respect to atmospheric concentrations for Isunnguata (685±230 ppm or 58±18 µM), near atmospheric equilibrium for Russell (442±31 ppm or 29±4 µM) and undersaturated for Kiattut Sermiat (263±33 ppm or 19±2 µM). $\delta^{13}C$-$CO_2$ values (Fig. 3h) are lower in spring and fall for Isunnguata (-16.6±4.0‰) compared to Russell (-13.7±2.3‰) and Kiattut Sermiat (-16.1±1.6‰) sites, though similar seasonal variation occurs for all sites with relatively more depleted values in the spring and fall compared to summer.



### 3.2 Methane oxidation and relationship with discharge

Values of $\varepsilon_c$ are similar throughout the melt season for Isunnguata ($38\pm10$‰) and Russell ($38\pm9$‰) and are relatively higher in the summer sampling period, while Kiattut Sermiat $\varepsilon_c$ values are higher on average ($42\pm13$‰; Fig. 4a) with lowest values in the summer. Estimates of $f_{ox}$ are similar between Isunnguata ($17\pm15\%$), Russell ($23\pm15\%$), and Kiattut Sermiat sites ($25\pm22\%$; Fig. 4b). However, $f_{ox}$ values are higher in the spring and fall sampling times compared to summer for Isunnguata and Russell and approach 50% in the spring, while Kiattut Sermiat values decrease throughout the melt season.

$CH_4$ concentrations are unrelated to Watson River average daily discharge for Isunnguata, but significantly negatively correlated for Russell (Fig. 5a). The fraction of $CH_4$ oxidized is moderately negatively correlated with discharge for both Isunnguata and Russell, though the relationship is not significant for either site (Fig. 5b). Discharge is positively correlated with $\varepsilon_c$ for both Isunnguata and Russell (Fig. 5c). While the relationship is only significant for Isunnguata, the slopes and intercepts of regression lines for Isunnguata and Russell are similar.

### 3.3 Mineral weathering and $CO_2$ models

Mineral weathering leads to net sequestration of $CO_2$ at all three sites (Fig. 6a). The magnitude of Net $\Delta CO_2$ differs between sites with average values lowest at Isunnguata ($-39\pm37$ µM) followed by Russell ($-65\pm32$ µM) and Kiattut Sermiat ($-98\pm17$ µM). Individual mineral weathering reactions produce differing contributions between sites and over the melt season, with notable differences between Watson River sites (Isunnguata and Russell) and Kiattut Sermiat (Fig. 6b). For instance, the proportional contribution of $Carb_{SA}$ is similar between Isunnguata ($17\pm11\%$) and Russell ($15\pm6\%$), though is lower at Kiattut Sermiat ($8\pm1\%$; Fig. 6b). Kiattut Sermiat has a relatively greater contribution from $Carb_{CA}$ ($62\pm2\%$) compared to Isunnguata ($41\pm10\%$) and Russell ($38\pm6\%$), while $Sil_{CA}$ is lower at Kiattut Sermiat ($28\pm1\%$) compared to Isunnguata ($41\pm17\%$), and Russell ($47\pm11\%$). Kiattut Sermiat additionally exhibits low seasonal variability in the proportional contributions of individual mineral weathering reactions compared to Isunnguata and Russell sites.

$CO_{2\text{-total}}$ represents $CO_2$ concentrations in the subglacial environment prior to addition and/or consumption of $CO_2$ through mineral weathering (Eq. 10; Fig. 7). Because the Net $CO_{2\text{-MW}}$ is always negative (more consumption then production), the value of $CO_{2\text{-total}}$ is always greater than measured concentrations ($CO_{2\text{-measured}}$). Regardless of differences in $CO_{2\text{-measured}}$ between sites, the average $CO_{2\text{-total}}$ values are similar between sites and average $91\pm47$ µM for Isunnguata, $94\pm33$ µM for Russell, and $117\pm16$ µM for Kiattut Sermiat.

Keeling plots between $[CO_{2\text{-total}}]^{-1}$ and $\delta^{13}C$-$CO_2$ indicate no linear relationship for Russell or Kiattut Sermiat samples, however a strong linear correlation is observed for Isunnguata ($r^2=0.99$; $p<0.001$) with the removal of one outlier, which also had the lowest $CO_{2\text{-total}}$ value (Fig 8a). Isunnguata samples also show a significant positive correlation between the magnitude of $CO_{2\text{-total}}$ and $NH_4$ concentrations ($r^2=0.73$; $p<0.05$), unlike Russell or Kiattut Sermiat samples, though several Russell samples fall close to the regression line between $CO_{2\text{-total}}$ and $NH_4$ of Isunnguata samples (Fig. 8b).



Because Isunnguata samples exhibit a linear relationship between $[CO_{2\text{-total}}]^{-1}$ and $\delta^{13}C\text{-}CO_2$ consistent with two endmember mixing, we utilize the $\delta^{13}C\text{-}CO_2$ of samples and end members defined by the Keeling plot in an isotopic mixing model to calculate the relative contributions of $CO_2$ sources to $CO_{2\text{-total}}$. We take the y-intercept (-27.5‰; Fig. 8a) of the Keeling plot regression as the $\delta^{13}C\text{-}CO_2$ for the high-$CO_2$ end member (assumed to be generated by OM remineralization; $CO_{2\text{-}OM}$) and assume atmospheric $CO_2$ as the low-$CO_2$ end member ($CO_{2\text{-atm}}$; $\delta^{13}C\text{-}CO_2 = -8$‰). The relative (%$CO_{2\text{-atm}}$ and %$CO_{2\text{-}OM}$) and absolute ($CO_{2\text{-atm}}$ and $CO_{2\text{-}OM}$) contributions of these $CO_2$ sources to Isunnguata $CO_{2\text{-total}}$ vary throughout the melt season, with $CO_{2\text{-}OM}$ being the dominant source in the early and late melt season while $CO_{2\text{-atm}}$ is the dominant source in the mid-melt season (Fig. 9a). The absolute magnitude of $CO_{2\text{-}OM}$ is approximately 5-10 times greater in the early (89 µM) and late (117 µM) melt seasons compared to mid-melt seasons (13-23 µM), while the magnitude of $CO_{2\text{-atm}}$ varies relatively little throughout the melt season at around 50 µM (Fig. 9b). Both %$CO_{2\text{-atm}}$ and %$CO_{2\text{-}OM}$ are significantly ($p<0.05$) correlated with Watson River discharge, and the correlation is positive for %$CO_{2\text{-atm}}$ and negative for %$CO_{2\text{-}OM}$ (Fig. 9c). The magnitude of $CO_{2\text{-}OM}$ exhibits a power-law relationship with discharge and is highest at lowest discharge, while the magnitude of $CO_{2\text{-atm}}$ is invariable as a function of discharge (Fig. 9d).

## 4 Discussion

We observe orders of magnitude variability in dissolved $CH_4$ and $CO_2$ concentrations in subglacial discharge of the GrIS, indicating significant differences in the magnitudes of the sources and sinks of these gases across time and space. Supersaturation of both $CO_2$ and $CH_4$ with respect to atmospheric concentrations indicates that Isunnguata discharge is a source of both gases to the atmosphere, neighboring Russell Glacier discharges water that is a source of $CH_4$ but near equilibrium with respect to $CO_2$, while Kiattut Sermiat in southern Greenland is a sink of atmospheric $CO_2$ but near equilibrium with respect to $CH_4$ (Fig. 3e, g). Because $CH_4$ dynamics may be largely microbially driven while $CO_2$ dynamics include microbial as well as abiotic mineral weathering processes, we first discuss $CH_4$ dynamics including a comparison of concentrations, isotopic compositions, and extent of oxidation between sites and over the melt season. We then discuss $CO_2$ concentrations, impacts of mineral weathering reactions (Table 1), and an assessment of subglacial $CO_2$ sources, including OM remineralization. These assessments will contribute to our understanding of the variability and controls of $CH_4$ and $CO_2$ concentrations in subglacial discharge from the GrIS and may improve predictions of the impact of future ice melt on Arctic carbon budgets.

## 4.1 Sources and sinks of CH₄

Differences in $CH_4$ concentrations and relationships with discharge between sites imply heterogeneity in both the extent and controls of subglacial methanogenesis under the GrIS. $CH_4$ supersaturation occurs at the two subglacial discharge sites that flow to the Watson River (Isunnguata and Russell), and concentrations are similar to the ranges reported in discharge of the Leverett Glacier (up to 600 nM; Lamarche-Gagnon et al., 2019), adjacent to the Russell Glacier in this study. However,





$CH_4$ concentrations are near atmospheric equilibrium for the Kiattut Sermiat site (Fig 2e). Because methanogenesis is an anaerobic OM remineralization pathway, it is more likely to occur in subglacial environments isolated from atmospheric $O_2$ sources. Widespread observations of methanogenesis in glacial meltwater of southwest Greenland from this and other studies

(Christiansen and Jørgensen, 2018; Dieser et al., 2014; Lamarche-Gagnon et al., 2019), and limited observations of $CH_4$ in subglacial discharge in southern Greenland, suggests heterogeneity in subglacial conditions that support methanogenesis. This heterogeneity may include variations in the distribution of subglacial organic substrates, such as subglacial permafrost reservoirs identified in previous studies (Ruskeeniemi et al., 2018), which were implicated as an OM source to drive methanogenesis under the Leverett Glacier (Lamarche-Gagnon et al., 2019). Organic deposits under the GrIS may have formed

as a consequence of Holocene ice margin fluctuations following the encroachment of ice over previously established soils and terrestrial vegetation (Ruskeeniemi et al., 2018). Methanogenesis fueled by organic material overridden during ice sheet growth has been suggested as a potential climate feedback over glacial interglacial timescales (Wadham et al., 2008), and may contribute to variations in $CH_4$ concentrations between southwest and southern sites in this study.

Subglacial methanogenesis may additionally be controlled by hydrologic factors as the subglacial hydrological

network develops throughout the melt season and channelization of meltwater conduits increases subglacial drainage efficiency (Andrews et al., 2015; Cowton et al., 2013). Drainage efficiency impacts both subglacial water residence time as well the transport of aerobic supraglacial meltwater to the ice bed. Both residence time and oxygen delivery may impact subglacial redox status and methanogenesis potential, and favor methanogenesis when oxygen supply rates are low compared to OM remineralization rates. This condition is most likely to be met in distributed subglacial systems that are hydrologically isolated

with limited inputs from aerobic supraglacial meltwater. Such a hydrologic control on methanogenesis at the Russell Glacier is supported by the significant negative correlation between $CH_4$ concentrations and Watson River average daily discharge (Fig. 5a), suggesting that $CH_4$ production occurs predominantly during periods of low discharge and greater residence time. Alternatively, higher $CH_4$ concentrations during low discharge could result from the dilution of relatively small volumes of methanogenic subglacial meltwater with increasing volumes of aerobic supraglacial meltwater. The lack of relationship

between $CH_4$ concentrations and discharge at the Isunnguata Glacier may indicate a greater influence of subglacial outburst events, in which hydrologically isolated methanogenic meltwater pockets are stochastically drained as the subglacial drainage network extends throughout the melt season (Fig. 5a). Subglacial outburst events were also implicated by heterogeneous $CH_4$ concentrations in subglacial discharge of the Leverett Glacier, and could contribute to heterogeneity in $CH_4$ concentrations at the Isunnguata Glacier (Lamarche-Gagnon et al., 2019).

While our results suggest heterogeneity in the extent and controls of methanogenesis between outlet glaciers, the microbial methanogenesis pathway as well as $CH_4$ oxidation dynamics are consistent between sites. Methanogenesis pathways may be evaluated by $\delta^{13}C\text{-}CH_4$ as well as $\varepsilon_c$ values because methanogenesis pathways impart different isotopic signatures to $CH_4$ and $CO_2$ (Whiticar and Schoell, 1986). Dieser et al. (2014) measured a microbial $\delta^{13}C\text{-}CH_4$ production signal at the Russell Glacier with values between -63‰ and -64‰, which was interpreted to reflect a possible combination of $CH_4$ produced through





both acetoclastic and $CO_2$ reduction pathways. The most depleted $\delta^{13}C$-$CH_4$ value measured at the Isunnguata in this study is close to that of Dieser et al. (2014) at -62.7‰ (Fig. 3f), and similar to values reported by Lamarche-Gagnon et al. (2019) for the Leverett Glacier, suggesting similar methanogenesis pathways across this region. While the exact contributions from each methanogenesis pathway cannot be inferred from isotopic information alone, the range of $\varepsilon_c$ values at outlet glaciers are consistent with predominantly acetoclastic methanogenesis during the peak melt season (Fig. 4a). However, $\varepsilon_c$ values fall

below the expected range from acetoclastic methanogenesis during the early and late melt seasons, likely resulting from variations in the extent of subglacial $CH_4$ oxidation. Seasonal variation in $CH_4$ oxidation is supported by consistency between $\varepsilon_c$ and $f_{ox}$ values, which both indicate the greatest impact of oxidation (approaching 50%) in the early melt season compared to peak melt season (Fig. 4a, b), with additional evidence of elevated $CH_4$ oxidation in the late melt season at both Isunnguata and Russell glaciers.

The extent of $CH_4$ oxidation may be controlled by multiple factors including oxygen availability, subglacial residences time, and subglacial hydrology, similar to methanogenesis. A hydrologic control of $CH_4$ oxidation is supported by relationships between $f_{ox}$ and $\varepsilon_c$ with Watson River daily discharge at both Isunnguata and Russell Glaciers: $f_{ox}$ is negatively related with discharge for both sites (Fig. 5b) while $\varepsilon_c$ is positively correlated with discharge (Fig. 5c) although the correlation is only significant at Isunnguata. These correlations suggest that $CH_4$ oxidation is greatest during periods of low flow, which

may be associated with greater residence times to allow subglacial $CH_4$ oxidation. The delivery of oxygen to the subsurface by supraglacial melting does not appear to be a limiting factor in subglacial $CH_4$ oxidation, which should increase $f_{ox}$ as more oxygenated supraglacial water is delivered to the subglacial system. Instead, the observed greater $CH_4$ oxidation during periods of low discharge may reflect $CH_4$ oxidation following mixing between draining methanogenic subglacial meltwater pockets with aerobic subglacial meltwater. Longer transit times during periods of low flow may allow more subglacial methane

oxidation to occur than during peak discharge, when the development of channelized flow paths reduces meltwater residence time in the subglacial environment.

Our results indicate a high degree of heterogeneity in subglacial methanogenesis under the GrIS, as well as a significant impact of $CH_4$ oxidation, which serves to reduce atmospheric $CH_4$ fluxes. Given the observed heterogeneity in this study, further investigation of the spatial variability in outlet glacier $CH_4$ concentrations is needed to determine the impact of

GrIS loss on Arctic and global $CH_4$ budgets, while a better understanding of the controls of these differences will improve models of how $CH_4$ fluxes from subglacial discharge will change with continued warming.

**4.2 Sources and sinks of $CO_2$**

Dissolved $CO_2$ concentrations in subglacial discharge are consistently supersaturated with respect to atmospheric concentrations at Isunnguata Glacier, near atmospheric equilibrium at Russell Glacier, and undersaturated at Kiattut Sermiat

Glacier, indicating that glacial meltwater from the GrIS can serve as either a source or sink of $CO_2$ to the atmosphere. Differences in dissolved $CO_2$ concentrations also imply variability in carbon processes under the GrIS (Fig. 3g). Because our





approach to assess the magnitude of subglacial $CO_2$ sources (including subglacial OM remineralization) depends in part on modelling results of $CO_2$ consumption by mineral weathering (Eq. 10), we first discuss the impacts of mineral weathering reactions, followed by a discussion of subglacial $CO_2$ sources, including OM remineralization.

### 375 4.2.1 Subglacial $CO_2$ sink: mineral weathering reactions

Although mineral weathering reactions may either increase or decrease dissolved $CO_2$ concentrations (Table 1), the net impact of mineral weathering at our study sites is to consume $CO_2$ (Fig. 6a). Net consumption occurs because the $CO_2$ source from $Carb_{SA}$ is ubiquitously low compared to sinks from either $Carb_{CA}$ or $Sil_{CA}$ (Fig. 6b). The range in Net $CO_{2\text{-}MW}$ is similar between subglacial discharge sites (between 10-150 µM; Fig. 6a), but average values increase from Kiattut Sermiat to
Russell to Insunnguata, likely reflecting the relative weatherability of alkaline igneous rocks, granulite facies gneisses, and amphibolite facies gneisses. Kiattut Sermiat is characterized by a relatively high proportion of $Carb_{CA}$ compared to Watson River sites, which may arise from the presence of trace carbonates in abundant readily weatherable basaltic intrusions as has been implicated in other studies (Urra et al., 2019). The relatively greater influence of carbonate dissolution compared to silicate dissolution on Total $CO_{2\text{-}MW}$ at Kiattut Sermiat may also relate to more rapid dissolution kinetics of carbonates, which
allow carbonate dissolution to have a large influence on major cation and anion load even when carbonates are only present in trace amounts (Tranter, 2005). At Isunnguata and Russell glaciers, $Sil_{CA}$ has a greater influence than $Carb_{CA}$ on Total $CO_{2\text{-}MW}$, which could result from either a lower abundance of trace carbonates to participate in weathering reactions, or relatively longer subglacial residence times that would allow a greater accumulation of silicate weathering products.

Despite the high impact of $Carb_{CA}$ on Total $CO_{2\text{-}MW}$ at Kiattut Sermiat compared to Isunnguata and Russell sites,
$Carb_{SA}$ is notably lower at Kiattut Sermiat than other sites and suggests a limited role for sulfuric acid weathering that may relate to subglacial sulfide oxidation dynamics. Lower abundances of sulfide minerals in the subglacial environment may limit the production of sulfuric acid, and could result from differences in lithology between sites, the depletion of sulfide minerals due to prior weathering (Graly et al., 2014), or weathering occurring in anoxic environments that limit the oxidation of sulfide to sulfuric acid (Deuerling et al., 2019). The kinetics of sulfide oxidation may also significantly differ between sites depending
on the relative contributions of abiotic compared to microbially mediated sulfide oxidation, as microbially mediated sulfide oxidation is several orders of magnitude faster than abiotic sulfide oxidation. Rapid microbially mediated sulfide oxidation has been implicated in the development of anaerobic conditions, which could also support subglacial methanogenesis (Sharp et al., 1999). Observations of higher $CH_4$ concentrations as well as higher contributions of $Carb_{SA}$ at Isunnguata and Russell compared to Kiattut Sermiat in this study may therefore be linked to subglacial microbial activity, which is known to vary
based on factors such as the presence of organic and fine-grained rock flour to serve as growth substrates, insulation from fluctuations in temperature, and delivery of nutrients and organic matter from supraglacial sources (Sharp et al., 1999). If microbially driven, our results suggest possible linkages between microbial processes and subglacial mineral weathering regimes, with significant impacts to both $CH_4$ and $CO_2$ dynamics due to the role of $Carb_{SA}$ as a $CO_2$ source (Table 1).



### 4.2.2 Subglacial $CO_2$ sources

Mineral weathering leads to net $CO_2$ consumption in all subglacial discharge samples, and thus the measured $CO_2$ concentrations represents only a fraction of the total $CO_2$ that would have been present in the absence of mineral weathering reactions ($CO_{2\text{-total}}$; Eq. 10). $CO_2$ sources could include dissolution of atmospheric gases in air-filled conduits or fractures in ice, or $CO_2$ contained in ice bubbles (Fig. 1; Anklin et al., 1995; Graly et al., 2017). $CO_2$ may also be produced through mechanical grinding and volatilization of fluid inclusions (Macdonald et al., 2018) or OM remineralization. While previous

studies have indicated that additional atmospheric $CO_2$ input through fractures and air-filled conduits may supply sufficient $CO_2$ to drive the observed extent of mineral weathering in many subglacial environments, including several sites in Greenland (Graly et al., 2017), $CH_4$ concentrations elevated above atmospheric equilibrium at the two Watson River sites reflects OM remineralization that would also contribute $CO_2$. While the magnitude of this source and its relative importance compared to other subglacial $CO_2$ sources is currently unknown, differing sources of carbonic acid for mineral weathering reactions carry

different implications for subglacial $CO_2$ budgets. For instance, carbonic acid weathering driven by invasion of atmospheric $CO_2$ would represent a sink of atmospheric $CO_2$, but carbonic acid weathering driven by OM remineralization would instead serve to consume $CO_2$ from *in situ* sources and limit its potential as an atmospheric source. Determining the sources of carbonic acid to subglacial weathering reactions is therefore critical to understand the controls of mineral weathering in subglacial environments as well as the role of that process in atmospheric $CO_2$ sequestration.

Comparisons between measured $\delta^{13}C$-$CO_2$ in subglacial discharge samples and likely $\delta^{13}C$-$CO_2$ values of $CO_2$ sources indicate that $CO_2$ sources differ between sites, with OM remineralization as the predominant $CO_2$ source at the Isunnguata but not at Russell or Kiattut Sermiat glaciers. Keeling plots of $[CO_{2\text{-total}}]^{-1}$ versus $\delta^{13}C$-$CO_2$ indicate that $CO_{2\text{-total}}$ at Isunnguata discharge may be represented by a two-end member mixing model, in contrast to Russell and Kiattut Sermiat glaciers (Fig. 8). Mixing model end members include a $^{13}C$-enriched, lower-$CO_2$ source and a $^{13}C$-depleted, higher-$CO_2$ source (Fig. 8a). The

y-intercept of the regression between $[CO_{2\text{-total}}]^{-1}$ versus $\delta^{13}C$-$CO_2$ (representing the isotopic signature of the high-$CO_2$ endmember) is -27.4‰, which is close to what would be expected from OM remineralization. For instance, $CO_2$ from remineralized OM in Greenlandic heath soils ranged between approximately -27 to -25‰ (Ravn et al., 2020), and between -20 to -30‰ for thawed Alaskan permafrost soils (Mauritz et al., 2019), both of which may be similar to subglacial organic matter. An additional correlation is observed between $CO_{2\text{-total}}$ and $NH_4$ concentrations for Isunnguata samples, which would

be expected from OM remineralization (Fig. 8b). The low-$CO_2$ end member could reflect atmospheric $CO_2$ input, which should result in a $\delta^{13}C$-$CO_2$ value of approximately -8‰. While the $\delta^{13}C$-$CO_2$ value of the lowest-$CO_{2\text{-total}}$ samples in the Isunnguata Keeling plot (e.g. highest $[CO_{2\text{-total}}]^{-1}$ not including the outlier) are slightly depleted compared to atmospheric values at -12.1‰ (Fig. 8a), even the lowest $CO_2$ concentrations measured at Isunnguata are supersaturated with respect to atmospheric concentrations (Fig. 3g). Supersaturation suggests that OM remineralization contributes $CO_2$ even for low $CO_2$-concentration

samples and isotopically depletes the subglacial $CO_2$ reservoir.





While both atmospheric $CO_2$ ($CO_{2\text{-atm}}$) and $CO_2$ derived from OM remineralization ($CO_{2\text{-OM}}$) provide carbonic acid to drive subglacial mineral weathering as well as $CO_2$ supersaturation in Isunnguata discharge, their absolute and relative contributions are controlled by different processes. Understanding the controls of $CO_2$ acquisition may improve understanding of subglacial carbon dynamics as well as the conditions necessary for subglacial environments to become $CO_2$ sources to the

atmosphere. $CO_{2\text{-OM}}$ is the dominant source to $CO_{2\text{-total}}$ at Isunnguata in the early and late melt seasons (Fig. 9a) when discharge is low (Fig. 9c), while $CO_{2\text{-atm}}$ is the dominant $CO_2$ source in the peak melt season when discharge is high. This switch in dominant $CO_2$ sources occurs because the magnitude of $CO_{2\text{-OM}}$ has a strong negative association with discharge, approaching zero during maximum discharge times (Fig. 9d), while $CO_{2\text{-atm}}$ remains relatively constant over the melt season (Fig. 9b) and the range of discharges (Fig. 9d). The high contributions of $CO_{2\text{-OM}}$ during low discharge could reflect higher residence times

that allow greater biogeochemical modification and accumulation of OM remineralization reaction products, including $CO_2$. The chemostatic behavior for $CO_{2\text{-atm}}$ indicates that invasion of atmospheric $CO_2$ is independent of the extent of chemical weathering, which exhibits strong seasonal variation (Fig. 6a). Chemostatic behavior of $CO_{2\text{-atm}}$ may indicate that $CO_{2\text{-OM}}$ maintains $CO_2$ concentrations at or above atmospheric saturation concentrations, and no additional atmospheric $CO_2$ dissolution would be needed to maintain equilibrium. Chemostatic behavior of $CO_{2\text{-atm}}$ could additionally indicate that the

Isunnguata subglacial drainage is largely closed to atmospheric exchange. Both mechanisms are supported by the consistent $CO_2$ supersaturation observed in subglacial discharge at Isunnguata (Fig. 3g), which suggests that limited atmospheric exchange prevents significant outgassing prior to discharge. These results first imply that $CO_2$ supersaturation due to OM remineralization is likely during low flow conditions in systems that are relatively closed to the atmosphere. Moreover, $CO_{2\text{-OM}}$ is the main driver for mineral weathering during low-flow conditions: while $CO_{2\text{-MW}}$ exceeds -150 μM late in the melt

season (Fig. 6a), $CO_{2\text{-atm}}$ provides only about a third of this $CO_2$ (Fig.8b), suggesting that the remainder was driven by $CO_{2\text{-OM}}$. In this case, only a fraction of the mineral weathering products measured in subglacial outflow at Isunnguata are directly involved in the sequestration of atmospheric $CO_2$, and the majority of mineral weathering serves to consume $CO_2$ from *in situ* $CO_{2\text{-OM}}$ production.

While $\delta^{13}$C-$CO_2$ values of Russell and Kiattut Sermiat samples are within the range of Isunnguata samples, suggesting

possible contributions of $CO_{2\text{-atm}}$ and $CO_{2\text{-OM}}$, non-linear Keeling plots indicate variability in the $CO_2$ concentration and/or isotopic composition of end members, or significant contributions of at least one other major subglacial $CO_2$ source. We address both possibilities here. While atmospheric $CO_2$ should be relatively invariable, $CO_{2\text{-OM}}$ may vary both in concentration and isotopic composition, depending on variability in the quantity and composition of organic deposits, as well as remineralization rates. For instance, if remineralization largely occurs in hydrologically isolated subglacial meltwater pockets,

some variability in the concentration and $\delta^{13}$C-$CO_2$ of $CO_{2\text{-OM}}$ is likely. While no data yet exist to characterize the variability in subglacial OM reservoirs, variability in either concentration or isotopic composition of $CO_{2\text{-OM}}$ could plausibly result in non-linear Keeling plots here.



Additional subglacial $CO_2$ sources could include atmospheric $CO_2$ contained in ice bubbles, or lithogenic $CO_2$ liberated by mechanical grinding, though both of these sources would be expected to enrich rather than deplete the $\delta^{13}C$-$CO_2$

values of the samples relative to modern atmospheric $\delta^{13}C$-$CO_2$ values. Ice bubbles contain gaseous $CO_2$ at concentrations and isotopic compositions reflecting atmospheric conditions during ice formation. While heterogeneity may result from gas bubbles recording changes in atmospheric $CO_2$, variability in $\delta^{13}C$-$CO_2$ of gas bubble $CO_2$ should be only a few per mil, which is small compared to the variation observed in Russell and Kiattut Sermiat samples (Tipple et al., 2010; Fig. 8a). Gas bubble $CO_2$ should also be $^{13}C$-enriched compared to modern atmospheric $CO_2$ due to fossil fuel contributions, and thus would be

unlikely to cause the variation in sample $\delta^{13}C$-$CO_2$ values that are more $^{13}C$-depleted than modern atmospheric $\delta^{13}C$-$CO_2$ values (Fig. 8a). Recent work has also highlighted the potential for subglacial mechanical grinding to result in $CO_2$ production through the volatilization of fluid inclusions (Macdonald et al., 2018). While volatilization of fluid inclusions through mechanical grinding was found to produce sufficient $CO_2$ to drive approximately 20% of mineral weathering in Svalbard subglacial environments, the expected isotopic composition of lithogenic $CO_2$ is more $^{13}C$-enriched than our measured $\delta^{13}C$-

$CO_2$ values. Because mechanical grinding should produce $CO_2$ with an isotopic composition reflecting the lithogenic source, (Lüders et al., 2012), its contributions here are likely limited. For instance, estimates of $\delta^{13}C$ for bulk hydrocarbons in fluid inclusions in the Ilímaussaq alkaline complex of South Greenland have values of -4.5±1.5‰ (Madsen, 2001), which is close to the $\delta^{13}C$-$CO_2$ of $CO_2$ in fluid inclusions in the Bamble granulite sector of South Norway (~ -6‰; Newton et al., 1980). There is an additional possibility of atmospheric exchange between the subglacial outlet site and our water sampling sites that

could serve as an additional $CO_2$ source or sink. However, atmospheric $CO_2$ exchange after discharge would have the same impact on Keeling plots as atmospheric $CO_2$ exchange prior to discharge. Although Kiattut Sermiat $CO_2$ concentrations are undersaturated with respect to atmospheric concentrations and would promote invasion of atmospheric $CO_2$, measured $\delta^{13}C$-$CO_2$ values are more $^{13}C$-depleted than modern atmospheric $CO_2$ and are not consistent with atmospheric $CO_2$ as the sole or dominant source of $CO_2$ to glacial meltwater samples (Fig. 8a). While more information is needed to determine the sources of

$CO_2$ to Russell and Kiattut Sermiat samples, $\delta^{13}C$-$CO_2$ values of samples from both sites imply mixing between a $^{13}C$-depleted $CO_2$ source, such as OM remineralization, and one or more $^{13}C$-enriched $CO_2$ sources, such as atmospheric or lithogenic $CO_2$.

## 5 Conclusions

Subglacial reactions impact the concentrations of $CO_2$ and $CH_4$ in subglacial discharge of GrIS, which act as either sources or sinks of GHG to the atmosphere. $CH_4$ concentrations of subglacial discharge are likely controlled by the availability

of subglacial OM to drive methanogenesis as well as the extent of $CH_4$ oxidation. Regional differences in subglacial OM deposits may account for the occurrence of methanogenesis in southwest outlet glaciers (Isunnguata and Russell) in this and other studies, contrasting with southern Kiattut Sermiat where little $CH_4$ production occurs. During the early melt season, oxidation consumes nearly 50% of $CH_4$ produced at southwest sites, and relationships between discharge and $CH_4$ oxidation





($\epsilon_c$ and $f_{ox}$) suggest that $CH_4$ oxidation depends on longer subglacial residence time during periods of low discharge. While mineral weathering consumes $CO_2$ throughout the melt season at all three sites, additional $CO_2$ resupplied from atmospheric and subglacial sources increases the $CO_2$ concentrations of subglacial discharge. The magnitude of additional $CO_2$ sources ($CO_{2\text{-total}}$) is insufficient to maintain atmospheric equilibrium at Kiattut Sermiat, leading subglacial discharge to be a sink of atmospheric $CO_2$, while $CO_{2\text{-total}}$ maintains close to atmospheric equilibrium concentrations at the Russell Glacier. At Isunnguata, however, OM remineralization produces more $CO_2$ than is consumed by mineral weathering and causes meltwater to be a source of $CO_2$ to the atmosphere. This finding implies that subglacial mineral weathering serves to partially or fully consume $CO_2$ produced from *in situ* sources under the GrIS but does not necessarily result in direct consumption of modern atmospheric $CO_2$. The important role of OM remineralization in subglacial environments of the GrIS determined by this and other studies also implies links between subglacial OM deposits and export of other biogeochemical solutes from the GrIS, including nutrients as well as redox-sensitive elements. While the export of nutrients from the GrIS has been the focus of numerous studies (Bhatia et al., 2013; Hawkings et al., 2016; Lawson et al., 2014), little is currently known regarding the role of OM sources in governing these exports. Given the variability in GHG concentrations observed in this study, constraining the extent of heterogeneity in outlet glaciers of the GrIS as well as the biogeochemical, hydrologic, and geologic controls of this heterogeneity will be important for upscaling atmospheric fluxes as well as efforts to predict impacts of ice loss on carbon budgets due to current and future melting of the GrIS.

**Author Contribution**

Jonathan B. Martin and Ellen E. Martin participated in conceptualization, data collection, data interpretation, reviewing and editing the manuscript, and acquired the funding for this project. Shaily Rahman participated in data collection and early interpretation and presentation of results. Andrea J. Pain participated in conceptualization, data collection, analysis, and interpretation, and took the lead on writing the manuscript with contributions from Jonathan B. Martin and Ellen E. Martin.

**Acknowledgements**

We acknowledge the members of our field teams: Daniel Fischer, Fabio Da Prat, Hailey Hall, Mark Robbins, and Scott Schnur. Additional invaluable support was provided by Steven DiEgidio, Nini Frydkjær Brandt, Inga Gisladottir, and Jacky Simoud. We are grateful for the excellent field support provided by the Kangerlussuaq International Science Station and Polar Field Services (CH2M Hill). This work was funded by the National Science Foundation grant (ANS-1603452). Discharge data from Watson River were gathered by B. Hasholt and A.B. Mikkelsen on behalf of the University of Copenhagen (2006-2013), and by D. van As and B. Hasholt on behalf of the Geological Survey of Denmark and Greenland (2014-present). The authors declare that they have no conflict of interest. Data is accessible on the Arctic Data Center (doi:10.18739/A2F76672G) including gas and nutrient data (https://cn.dataone.org/cn/v2/resolve/urn:uuid:c1051a07-cbdf-4061-ae44-c1472f61e3fe) and major



element concentrations used for geochemical modeling (https://cn.dataone.org/cn/v2/resolve/urn:uuid:65d272f6-d280-4fcc-
8aaa-4805f12ca6ae).

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





**Table 1. Mineral weathering reactions and impacts on dissolved $CO_2$ concentrations.**

| Eq. | Mineral | Acid | Abbreviation[1] | Reaction | Impact on $CO_2$ |
|-----|---------|------|-----------------|----------|------------------|
| (1) | Carbonate | Carbonic | Carb$_{CA}$ | $(Ca, Mg)CO_3 + H_2O + \mathbf{CO_2} \rightarrow (Ca^{2+}, Mg^{2+}) + 2HCO_3^-$ | $CO_2$ sink |
| (2) | | Sulfuric | Carb$_{SA}$ | $2(Ca, Mg)CO_3 + H_2SO_4 \rightarrow 2(Ca^{2+}, Mg^{2+}) + SO_4^{-2} + H_2O + \mathbf{CO_2}$ | $CO_2$ source |
| (3a) | Silicate | Carbonic | Sil$_{CA}$ | $(Ca, Mg)Al_2Si_2O_8 + \mathbf{2CO_2} + 3H_2O \rightarrow (Ca^{2+}, Mg^{2+}) + 2HCO_3^- + Al_2Si_2O_5(OH)_4$ | $CO_2$ sink |
| (3b) | | | | $(Na, K)AlSi_3O_8 + \mathbf{CO_2} + 5.5H_2O \rightarrow (Na, K) + HCO_3^- + 0.5Al_2Si_2O_5(OH)_4 + 2H_4SiO_4$ | $CO_2$ sink |
| (4a) | | Sulfuric | Sil$_{SA}$ | $(Ca, Mg)Al_2Si_2O_8 + H_2SO_4 + H_2O \rightarrow (Ca^{2+}, Mg^{2+}) + SO_4^{-2} + Al_2Si_2O_5(OH)_4$ | No impact |
| (4b) | | | | $2(Na, K)AlSi_3O_8 + H_2SO_4 + 9H_2O \rightarrow 2(Na^+, K^+) + SO_4^{2-} + Al_2Si_2O_5(OH)_4$ | No impact |

[1] Abbreviations are based first on the mineral class (carbonate = carb, silicate = sil) and then on the acid (carbonic acid = CA, sulfuric acid = SA)





**Figure 1: Conceptual diagram of subglacial sources and sinks of $CO_2$ and $CH_4$. Arrow indicate the direction of fluxes. Boxes represent processes, and sources of gases to subglacial meltwaters are indicated by green text while sinks of gases to subglacial meltwater are indicated by red text. Gas bubbles, mechanical grinding, and OM remineralization are grouped because all are $CO_2$ and $CH_4$ sources.**





**Figure 2.** © Google Earth satellite images of study locations in a) Greenland including b) Isunnguata (dark blue circle) and Russell
(light blue square) subglacial discharge sites flowing to the Watson River in southwest Greenland (c) and the Kiattut Sermiat site
(orange triangle) in southern Greenland. Discharge monitoring of the Watson River is collected by PROMICE (van As et al., 2018)
at the location represented by the yellow star.




**Figure 3. Chemical parameters at Isunnguata (IS), Russell (RU) and Kiattut Sermiat (KS) subglacial discharge sites versus day of year for a) specific conductivity (Sp.C), b) pH, c) dissolved oxygen (D.O.) percent saturation, d) alkalinity (Alk), e) measured CH₄ concentrations (left y-axis in ppm and right y-axis in nM), f) δ¹³-CH₄ values, g) measured CO₂ concentrations (left y-axis in ppm and right y-axis in μM), and h) δ¹³C-CO₂ values. Atmospheric equilibrium concentrations are indicated by dashed lines and taken as 1.9 ppm for CH₄ and 410 ppm for CO₂. Error bars on CH₄ and CO₂ concentrations and stable isotopic compositions represents the standard deviation of replicates and are smaller than symbols for some data points.**





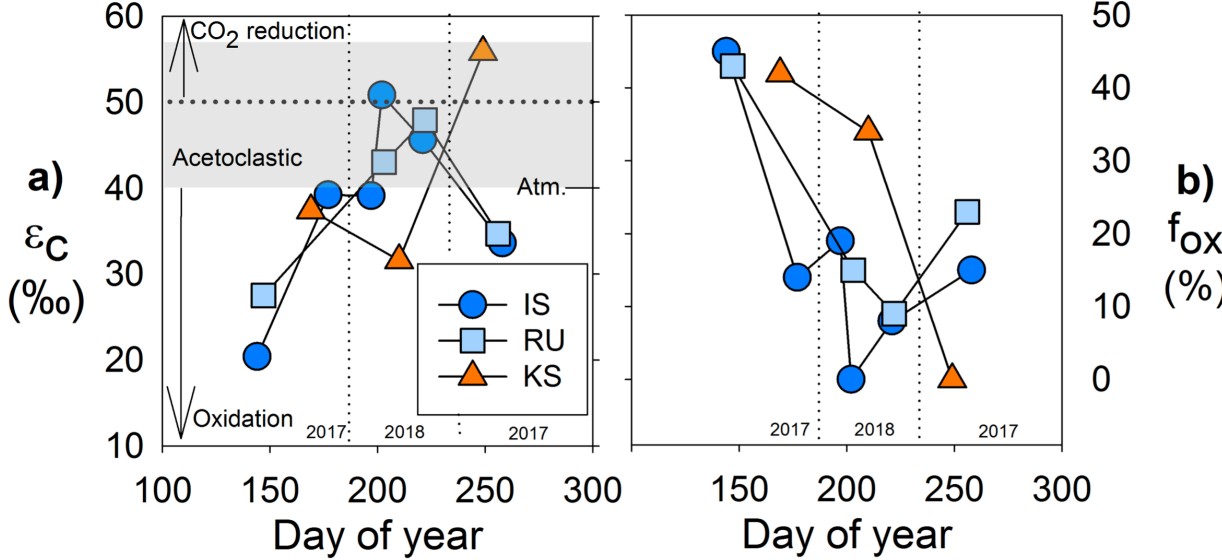

**Figure 4. CH₄ dynamics over the course of the 2017 and 2018 melt seasons including a) the carbon fractionation factor (εc) between dissolved CO₂ and CH₄ and b) the fraction of CH₄ oxidized (fox) for Isunnguata (IS), Russell (RU) and Kiattut Sermiat (KS) samples. Fields of εc representing methanogenesis and oxidation values are based on Whiticar (1999). Values of εc between approximately 40 and 55 are produced for methanogenesis via acetate fermentation, while CO₂ reduction produces values between approximately 50 and 90. Lower values result from a predominant isotopic signature of CH₄ oxidation. Atmospheric input without additional alteration of CO₂ or CH₄ isotopic systematics results in a εc value of approximately 40.**

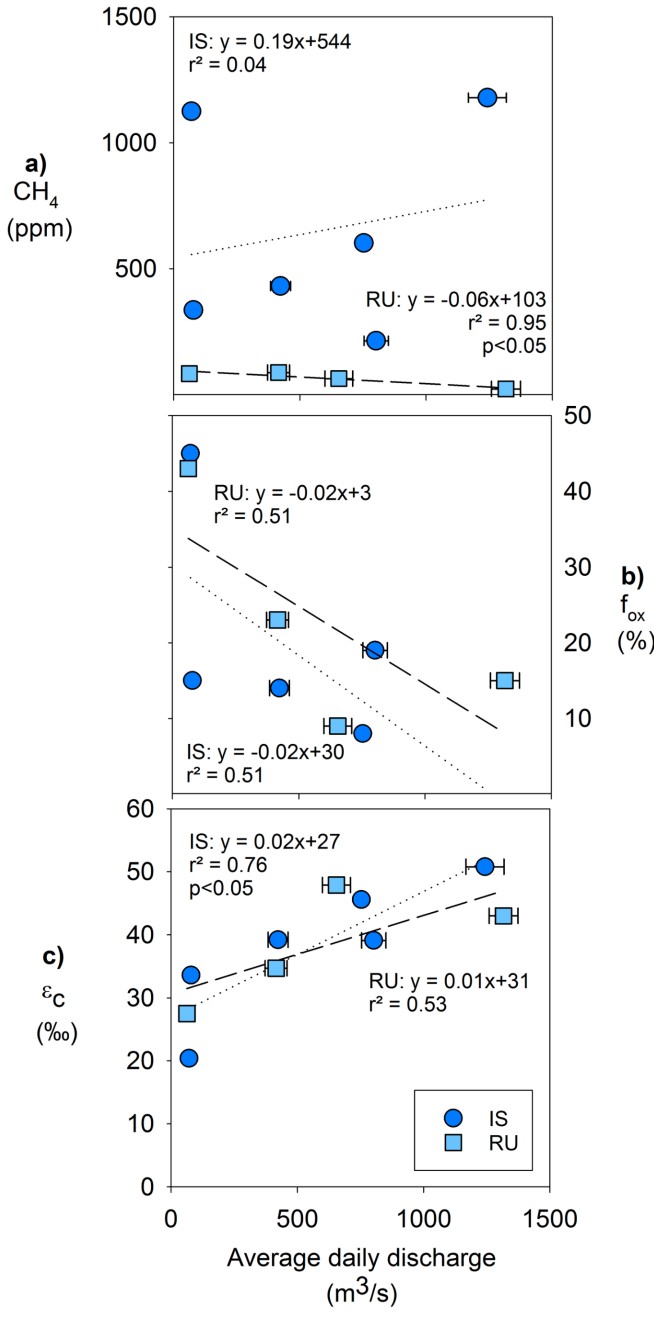

**Figure 5. Discharge relationships with CH₄ dynamics including a) CH₄ concentrations, b) f_ox, and c) ε_c for Isunnguata (IS) and Russell (RU) samples. Regressions are shown by dotted lines for Isunnguata and dashed lines for Russell samples. Average daily discharge estimates are for the outlet of the Watson River and provided by PROMICE (van As et al., 2018). Horizontal error bars represent the standard deviation of average daily discharge for days samples were collected and are smaller than symbols for some data points.**




**Figure 6.** Mineral weathering model results in a) net impact of mineral weathering reactions on CO₂ (Net CO$_{2\text{-MW}}$; Eq. 7) for
Isunnguata, Russell, and Kiattut Sermiat subglacial discharge sites (where negative values of Net CO$_{2\text{-MW}}$ indicate net sequestration
of CO₂ due to mineral weathering), and b) the proportional contribution of each mineral weathering reaction to the total change in
CO₂ (% Total CO$_{2\text{-MW}}$ Eq. 9a-9c).


**Figure 7.** Calculated $CO_{2\text{-total}}$ values for a) Isunnguata, b) Russell, and c) Kiattut Sermiat subglacial discharge against day of the year.



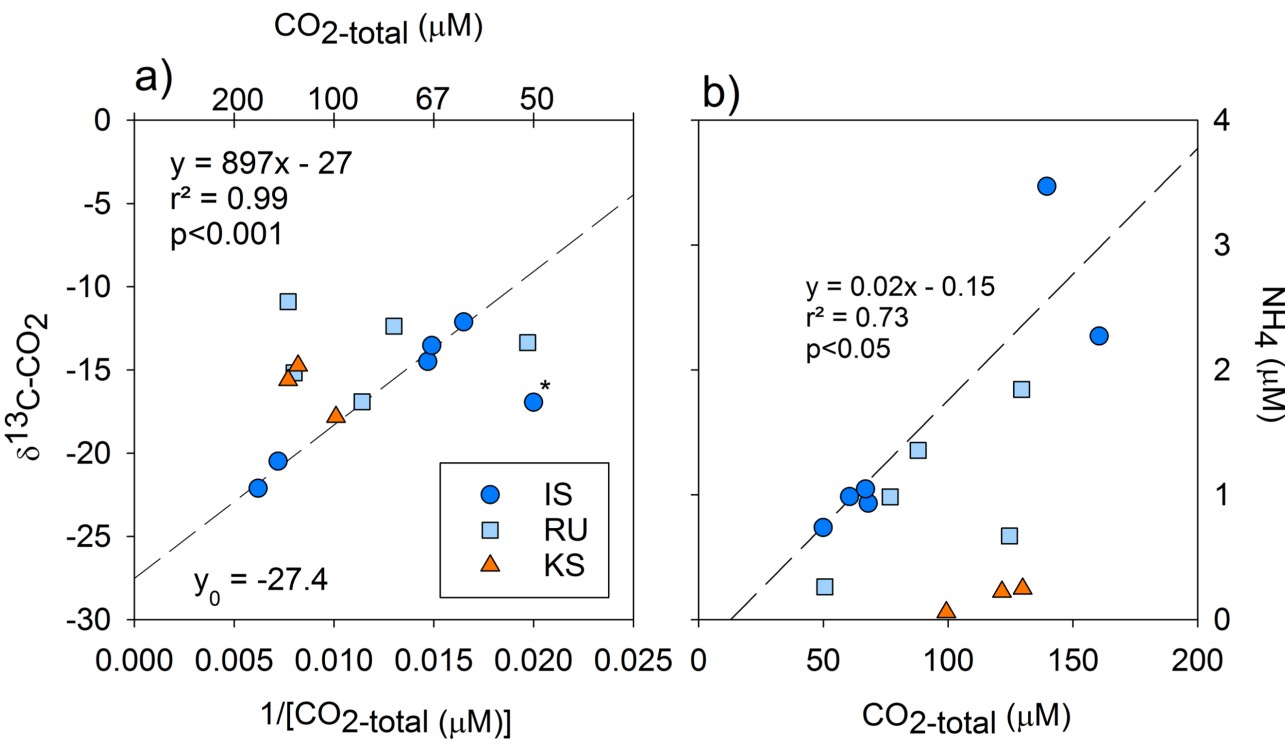

**Figure 8. Correlations between the magnitude of $CO_{2\text{-total}}$ and a) $\delta^{13}C\text{-}CO_2$ and b) $NH_4$ for Isunnguata (IS), Russell (RU) and Kiattut Sermiat (KS) samples. Asterisk denotes the Isunnguata outlier not included in the regression between $CO_{2\text{-total}}$ and $\delta^{13}C\text{-}CO_2$ in panel a).**





**Figure 9. Estimates of contributions of atmospheric $CO_2$ ($CO_{2\text{-atm}}$) compared to that derived from remineralized OM ($CO_{2\text{-OM}}$) for**
**Isunnguata subglacial discharge site including a) relative contributions according to the day of the year, b) absolute magnitudes**
**according to the day of the year, c) relative contributions according to Watson River average daily discharge, and d) absolute**
**magnitudes according to Watson River average daily discharge. Average daily discharge estimates are for the outlet of the Watson**
**River and provided by PROMICE (van As et al., 2018). Horizontal error bars represent the standard deviation of average daily**
**discharge for days samples were collected and are smaller than symbols for some data points. Regressions between average daily**
**discharge and proportional and absolute contributions of $CO_{2\text{-atm}}$ and $CO_{2\text{-OM}}$ sources are shown with dotted lines.**