# Peer review of "Heterogeneous CO2 and CH4 content of glacial meltwater from the Greenland Ice Sheet and implications for subglacial carbon processes"

_The Cryosphere, 2020_

## Short Comment (SC1) · 15 Jul 2020

A small comment about the place names used in this paper:

The authors refer to their northernmost site as "the Isunnguata glacier" and subsequently as simply "Isunnguata". The place name of the site is Isunnguata Sermia. Sermia meaning Glacier in Greenlandic. It is the same word that appears in their other Greenlandic place name Kiattut Sermiat, but in a different grammatical form. The authors should either include Sermia/Sermiat in both place names or drop it from both (as they have from Russell Glacier). However, Isunnguata or Kiattut are nonsensical phrases in Greenlandic on their own, so the former would be a better option. The

authors should also fix the spelling of Kiattut Sermiat in figure 2.

When discussing Isunnuguata Sermia, the authors should make it clear that they are discussing the glacier's southern lateral outlet, not the terminal outlet. The terminal outlet is connected to a deep subglacial trough and fed from an ∼16,000 kmˆ2 catchment, whereas the lateral outlet seems to have a much more local source. Without this distinction, readers may incorrectly relate your results to this larger catchment.
* * *

---

## Referee Comment (RC1) · Anonymous Referee #1 · 29 Jul 2020

GENERAL COMMENTS: Pain et al. report carbon dioxide (CO2) and methane (CH4) concentrations, 13C stable isotopes and water chemistry from subglacial meltwaters of three glaciers located in southern regions of Greenland. The authors do a good job of describing these systems and discussing controls on both CO2 and CH4 dynamics in glacial systems. The manuscript is generally well written and easy to follow. In my opinion, this is a worthwhile contribution to the literature on the biogeochemistry of glacial systems. My suggested changes are mostly technical, but I distinguish below between "Specific Comments", referring to systematic changes in multiple sections of the manuscript or more substantive changes, from "Technical Comments", which are editorial.

[Figure]

SPECIFIC COMMENTS: Could the novelty of the study be better highlighted? A significant body of work exists on the southwestern glaciers in particular (Innunguata, Russell), as described on lines 103-113, so it's not entirely clear to the reader what the novel contribution of this study is from the text at the outset.

Are errors throughout reported as standard deviations or standard errors (e.g., see section 3.1)?

The samples were collected over a couple of years, which is perfectly reasonable for Arctic sites given the finances and logistics of working in the region. However, since the samples collected in 2018 were from the summer, and the 2017 samples were from the spring and fall, it would seem inappropriate to display the points with adjoining lines as a time series (i.e., Figures 3,4, 6, 7), because there can be large interannual differences in meltwater dynamics. There is no perfect solution to this, except to remove the lines adjoining the 2017 and 2018 samples, and perhaps discuss differences between the two years given the DMI climate data for both regions and/or the PROMICE discharges for the two southwest glaciers.

The justification for the measurement and presentation of the NH4+ data aren't obvious (only stated on L430). This should be explicitly indicated in the methods, but the data are not particularly informative and could be excluded (though this is entirely up to the authors).

L271-275, Fig. 8: The statistics presented in the text vs. Fig. 8 are a bit confusing to follow. The text states that it's a correlation (i.e., independence of x and y), but a linear regression (i.e. dependence between x and y) is shown. Correlation statistics (i.e., r instead of r2) should be shown, or the text should be changed to reflect a linear regression (which is otherwise used throughout the text).

Further, from Fig. 8a, the statistics appear to apply to the entire dataset as there is no indication from the caption or figure otherwise; however, the text states that the relationship was only observed for the Isunnguata samples, suggesting that the statistics presented only refer a subset of the data presented. I also wonder about the validity of removing the outlier... It's possible that this relationship is not linear, but rather parabolic, with lower 13C values at lower CO2 concentrations indicative of rapid weathering, which can mimic the 13C-DIC signature associated with OM remineralization. There are not enough data to test this, but it would be something to keep in mind as the deviation of the "outlier" is not large enough to be indicative of analytical issues, but perhaps a true pattern. For this reason, I'm a little hesitant about the inference of a simple two end-member mixing model, especially since it does not seem to hold for the other sites.

TECHNICAL COMMENTS:

L26-27: Add reference to sentence starting with "Variations. . ." But also see Tranter et al. 2002, that has discounted a substantial role for glacial weathering on atmospheric CO2 concentrations over geological time scales (https://doi.org/10.1016/S0009-2541(02)00109-2).

L32-35: It might be useful to explain here the possible sources of CO2. As is, it seems somewhat disjointed from the preceding sentence, which describes CO2 budget. Both CO2 and CH4 will contribute to the carbon budgets of the system. The sources of CO2 are discussed in the following paragraph, but perhaps just a rejigging of this text would read more fluidly. One option would be to move L32-39 after the following paragraph and slightly expand upon the CH4 introduction before introducing the purpose of the paper. For example, of additional relevance to CH4 in subglacial environments is the formation of the necessary precursor H2 by rock comminution in Telling et al. 2015 (https://doi.org /10.1038/ngeo2533).

Section 2.1: What is the seasonality of these systems? How much of the annual discharge occurs during the period where these were sampled? Is there winter flow? Glacial outburst floods?

L103-113: See also Dubnick et al. (2017; https://doi.org/10.1002/2016JG003685),

which includes calculated $CO_2$ undersaturation at Kiattut Sermiat.

L115: What were the specific sampling dates? If this is too much detail to have as text, then at least the number of sampling campaigns at each site each year would be useful information here.

L240: What was the $13C$-$CO_2$ value for summer across all sites?

Figure 1: Subscripts indicating carbonic or sulphuric acid (CA/SA) should be defined in the caption or on the figure.

Figure 2b: I'm wondering if there would be a way to trace the Watson River. It is difficult to see how the two study glaciers feed into the river.

Figure 2c: Kiattut Sermiat is spelled differently (Kiagtut) in panel c than elsewhere in the manuscript.

Figure 7: This caption should be more informative so that the figure can stand alone without the text. The y-axis is not intuitive without the definition for $CO_2 total$ provided in the text (L266-267). As is, it's confusing because it looks like $CO_2$ concentrations are negative, which in principle is impossible, though I understand what the figure shows.

Figure 7: Could colour blocks instead of circle symbols be used for the legend? It's a small detail, but otherwise only technically refers to the Isunngata panel.

Figure 9: Symbology of the regression lines should be different from for the separation between years in which samples were collected.

---

## Referee Comment (RC2) · Marek Stibal (Referee) · 31 Jul 2020

**Review of 'Heterogenous CO₂ and CH₄ content of glacial meltwater of the Greenland Ice Sheet and implications for subglacial carbon processes'**

The authors measured $CO_2$ and $CH_4$ concentrations and C stable isotopic signatures in meltwater samples from three marginal catchments of the Greenland ice sheet (GrIS) to evaluate sources and sinks of these important greenhouse gases under the fast melting ice sheet. The study is timely, well-structured and -written, and uses adequate methods (with exceptions described below). However, interpretation of some of the results (especially the $CH_4$ part) relies on incorrect assumptions and/or is unsupported by data, and is therefore too speculative at best. The issues listed below need to be carefully addressed before the study can be published.

First, the authors misunderstand and/or misrepresent the regional hydrology of the Kangerlussuaq area. The large outlet Isunnguata Sermia drains into the Isortoq River/Isortup Kuua (N of the Watson River catchment); its catchment is indeed quite large and extends deep into the ice sheet (probably to the ice divide). However, the water samples collected in this study did not originate from this large catchment and using the name Isunnguata Sermia is erroneous. Rather, the authors' IS site falls into the Point660 subcatchment sensu Lindbäck et al 2015 (as the authors admit at line 88), with an area of ca 30-60 $km^2$. This is part of the Russell Glacier catchment system (sometimes treated as part of the Leverett Glacier catchment, eg in Lindbäck et al. 2015), which is complicated, but likely does not reach far into the ice sheet. Calculating the Russell catchment area (for the RU site) as the difference between two individual and independent estimates of the Leverett catchment is incorrect (for example, some authors estimated the Leverett catchment at >1000 $km^2$); the catchment feeding the Russell river is likely much smaller than 300 $km^2$ (see eg van de Wal and Russell 1994). Figure 2b attributes the name Watson River to the river system originating at Point 660 and joining the larger river discharging from Leverett Glacier, which is problematic. Whereas the river terminology in the area is indeed somewhat confused, Watson River usually refers to the last short section of the river formed by the confluence of Akuliarusiarsuup Kuua and Qinnguata Kuussua which then enters Kangerlussuaq where the hydrology data used in the study (from van As et al 2018) were taken.

Moreover, it is difficult to compare the study sites with previous works as the authors neither give details of their positions nor show any pictures. I assume the IS site is identical to the site used in Christiansen & Jørgensen (2018). The $CH_4$ supersaturation referred to at lines 107-108 was measured in the air, with respect to atmospheric concentration, not in the water. The stream itself is very small (discharge at the portal was ca 1 $m^3$/s at peak melt in July 2018) and partly/mostly of supraglacial origin (a supraglacial/marginal stream can be found flowing into the ice margin several hundred meters above the portal). In terms of the RU site, it is not clear whether the authors sampled the main stem of the Russell Glacier river or one of the short subglacial tributaries flowing into it from the Russell Glacier margin (as depicted eg in Dieser et al 2014). The Russell Glacier river in this area has already flowed through a number of lakes and the $CH_4$ signal would be difficult to interpret as purely subglacial. The order of magnitude difference in $CH_4$ concentrations reported here and in Dieser et al (2014) is not discussed in the text but suggests that indeed samples were collected from the main stem of the river. A better description of the sampling sites is essential for an adequate assessment of the authors' interpretation.

While the inaccuracies in the description of the regional hydrology are easy to fix, they led the authors to a more problematic aspect of the study: correlating the $CH_4$ and $CO_2$ concentrations and isotopic signals from the IS and RU sites with discharge data from the Watson River. While both the Russell Glacier and Watson rivers show clear diurnal variations in discharge, large scale dynamics (including subglacial outbursts) observed in the large rivers is missing in the smaller Russell Glacier river, possibly due to the buffering effect of the lakes along the course of the river, as illustrated in

the figure below comparing unpublished discharge data in m³/s from LG (Leverett Glacier river), RU (Russell Glacier river), and IS (Isunnguata Sermia river/Isortup Kuua) in the summer 2018. The Watson River discharge dynamics may be further complicated by the unaccounted for Qinnguata Kuussua, which provides more than half of its water. This is in contrary to the authors' assumptions (line 217).

[Figure]

To my knowledge, this dataset has been made available to the authors, and it might be beneficial for the authors to use it for their correlations. Maybe it was not used because there were just 2 and 3 samples collected at RU and IS, respectively, in 2018 (as shown in Figures 3, 4, 6, 7, 9)?

The low number of samples and the fact they come from two different years, 2017 and 2018, is another weakness of the study. Figures 3, 4, 6, 7, 9 appear as though they show time series (suggested by the lines connecting the dots), which is not the case. I suggest the authors redraw the figure so it's clear the data come from two independent seasons. This should also be acknowledged in the text, and the possible differences in hydrology and the potential impact on the export of gases discussed more in detail. For example, Hawkings et al (2015) showed large interannual variations in total discharge and solute and particulate fluxes from the Leverett catchment between 2009-2012. Also, no statistical analysis of the results was done and it's unclear whether the differences in water chemistry were significant between the streams – was this due to the low number of samples? This should also be acknowledged and/or explained.

Analysis of stable isotopic signatures of C in $CH_4$ and $CO_2$ is a powerful tool for determining the origin of the respective gases; however, caution must be exercised when interpreting the results for glacial meltwater samples. Glacial meltwater is a very dynamic mixture consisting of several components with different origins: the water mostly comes from the surface and so has been in direct contact with the atmosphere (and its $CO_2$); the sediment is predominantly of subglacial origin. Dissolved $CH_4$ also likely originates from the subglacial environment, while $CO_2$ has multiple sources -- as the authors show and discuss throughout the ms. The isotopic separation factor $\varepsilon_C$ (ie $\delta^{13}C_{CO2}$ - $\delta^{13}C_{CH4}$), used in this study to assess the sources and sinks of $CH_4$, is suitable for closed systems (as defined in Whiticar 1999), but caution must be exercised when using it for glacial meltwater and the limitations should be acknowledged and discussed in the text. The authors also calculate the fraction of oxidised methane ($f_{ox}$) using a number of assumptions, some of which might not be substantiated. For example, Michaud et al (2017) modelled the kinetic isotopic fractionation factor $\alpha_{ox}$ beneath the West Antarctic Ice Sheet, an environment likely to be more representative of the bed of the GrIS, at

1.004. The authors use a value of 1.049, which may lead to an underestimation of microbial oxidation of $CH_4$ in the GrIS subglacial system. More importantly, outgassing, as a major process affecting meltwater gas concentrations, should not be ignored. In the turbulent flow of glacial rivers, most $CH_4$ will likely outgas very quickly: for example, in the Leverett Glacier river, virtually all $CH_4$ is gone after ca 2 km (Lamarche-Gagnon et al 2019). Moreover, outgassing affects not only the concentrations, but also the isotopic composition of gases due to fractionation (see eg Banks et al 2017), driving the remaining dissolved gas to more positive (heavier) values. This may result in an overestimation of $CH_4$ oxidation. While outgassing was probably less significant at IS (as the authors sampled only 10 m from the subglacial outlet), it may have affected gas concentrations at RU (100 m) and definitely would have at KS (>1 km and a proglacial lake; see below). Much more attention should be paid to the possible effects of this process in the discussion. Moreover, it should be pointed out that outgassing is likely enhanced in glacial systems by considering the large pressure differentials between the subglacial environment where the $CH_4$ is produced and the atmosphere, and the rapid depressurisation that results from pressurised subglacial waters exiting the ice. Such depressurisation effect is likely to also influence the isotopic signature of the sampled gases in runoff (Banks et al 2017). While accounting for outgassing/depressurisation and their effects on isotopic fractionation might be difficult, if not impossible, the authors should at least discuss the limitations and biases of not doing so, and whether or not the assumptions from their oxidation model would still hold true.

The KS site is additionally problematic as there is a large proglacial lake right by the portal, with an estimated water residence time in the order of 24 hours at peak discharge (Hatton et al 2019). This may significantly change the concentrations and isotopic signatures of the dissolved gases exported further downstream via outgassing (and possibly also microbial processes in the lake bottom sediment), and may be the reason why the $CH_4$ concentrations at KS are near atmospheric equilibrium. This should also be mentioned when discussing the results from KS.

In the discussion, the authors interpret the observed orders of magnitude variability in dissolved gas concentrations in the meltwater samples as differences in the sources and sinks of the gases (lines 290-291). However, some of the explanations of $CH_4$ variability are unnecessarily speculative and unsupported by data, and some rest on incorrect assumptions. First, the variability in subglacial OM substrates is invoked (312). This is certainly a factor affecting subglacial C cycling and export rates, but no supporting OC data are presented. Permafrost reservoirs, suggested based on the study by Ruskeeniemi et al (2018), are unlikely to be of importance (and were not alluded to in Lamarche-Gagnon et al 2019, as suggested at line 314), as they extend only a few km into the ice sheet bed. Moreover, Ruskeeniemi et al (2018) only focused on the thermal state of the sediments/soils, rather than the nature of OC. I agree the Holocene ice margin fluctuations were probably important in providing fresh OC substrate that could have been metabolised into $CH_4$ that is currently being exported. Older (Eemian) OC sources are however also present and exported in the meltwater (Kohler et al 2017) and may have been used as methanogenesis substrates. Reservoirs of old $CH_4$ are not considered in the study. Second, a direct effect of oxygen supply to the ice sheet bed by surface meltwater on methane production/oxidation is proposed, based on the negative correlation of $CH_4$ concentrations at RU and Watson River discharge (319-327). As explained above, linking gas concentrations and isotopic signatures at IS and RU to discharge data from the Watson River is misleading. In addition, the authors only consider live methanogenesis and ignore potential old $CH_4$ storage/leakage (see above). Dilution by meltwater is only acknowledged at lines 328-329 as an alternative explanation, although it plays a significant role. The local subglacial sources of $CH_4$ are probably limited to microbial activity (Lamarche-Gagnon et al 2019), which takes place in anoxic sediments buried under the ice. Whether it's recent activity or reservoirs of ancient $CH_4$, its export is dependent on meltwater tapping and flushing pockets of produced gas. As a result, $CH_4$

concentrations in the meltwater are necessarily discharge-dependent. This is indeed complicated by outburst events; however, these are limited to large outlets (lakes form at much higher altitudes further into the ice sheet than those to which this subcatchment extends), and explaining the lack of discharge-$CH_4$ concentration relationship at IS by outbursts (330-332) is therefore is not justified. Last, $CH_4$ oxidation, discussed at lines 350-361, is certainly an important process controlling the amount of $CH_4$ that will be exported from under the ice to the atmosphere. However, in addition to the uncertainty in determining the degree of $CH_4$ oxidation, the authors' interpretation of the data again relies on correlating the $CH_4$ concentrations at IS and RU with Watson River discharge and on treating the 2017 and 2018 data as a time series, both of which are flawed (see above).

In summary, I recommend the authors revisit their local hydrology description and interpretation, rename their sampling sites accordingly, avoid correlating their small stream data with the Watson River discharge record, and properly acknowledge the limitations and uncertainties of the used geochemical calculations for interpretation of the subglacial gas sinks and sources, especially for $CH_4$.

**Specific comments**

53 please specify if Graly et al 2017a or b

58-60 relevant work should be cited here, eg the recent review by Wadham et al (2019)

66 Musilova et al (2017) did not study subglacial microbial activity; this reference is irrelevant here

107-110 methanogens have also been identified in Russell Glacier basal ice (Stibal et al 2012) and Leverett Glacier river suspended sediment (Lamarche-Gagnon 2019); $CH_4$ supersaturation in meltwater was also measured by Dieser et al (2014) but not by Christiansen & Jørgensen (2019)

306 Lamarche-Gagnon et al (2019) measured higher $CH_4$ concentrations than 600 nM (up to 4000 nM during early season)

405 how do the results compare to the recent paper by Andrews et al (2018) focused on dissolved C dynamics in Russell Glacier meltwater, including the sources of subglacial $CO_2$?

446 please explain 'chemostatic behavior'

695 Figure 1 is a weird combination of 2D and 3D which makes it difficult to interpret. Also, could the authors provide references for $CO_2$ and $CH_4$ evasion through crevasses?

700 Figure 2 needs redrawing to correct the river network names and to better indicate the sampling sites; please also use the newer transcription 'Kiattut', to be consistent with the text.

740 the regression line in Figure 8b doesn't look right – were some points omitted?

**References**

Andrews MG, Jacobson AD, Osburn MR, Flynn TM (2018) Dissolved carbon dynamics in meltwaters from the Russell Glacier, Greenland Ice Sheet. *J Geophys Res Biogeosci* 123:2922–2940

Banks EW, Smith SD, Hatch M, Burk L, Suckow A (2017) Sampling dissolved gases in groundwater at in situ pressure: a simple method for reducing uncertainty in hydrogeological studies of coal seam gas exploration. *Environ Sci Technol Lett* 4:535–539

Christiansen JR, Jørgensen CJ (2018) First observation of direct methane emission to the atmosphere from the subglacial domain of the Greenland Ice Sheet. *Sci Rep* 8:16623

Dieser M, Broemsen ELJE, Cameron KA, King GM, Achberger A, Choquette K, Hagedorn B, Sletten R, Junge K, Christner BC (2014) Molecular and biogeochemical evidence for methane cycling beneath the western margin of the Greenland Ice Sheet. *ISME J* 8:2305–2316

Hatton JE, Hendry KR, Hawkings JR, Wadham JL, Kohler TJ, Stibal M, Beaton AD, Bagshaw EA, Telling J (2019) Investigation of subglacial weathering under the Greenland Ice Sheet using silicon isotopes. *Geochim Cosmochim Acta* 247:191–206

Hawkings JR, Wadham JL, Tranter M, Lawson E, Sole A, Cowton T, Tedstone AJ, Bartholomew I, Nienow P, Chandler D, Telling J (2015) The effect of warming climate on nutrient and solute export from the Greenland Ice Sheet. *Geochem Persp Lett* 1:94-104

Kohler TJ, Žárský JD, Yde JC, Lamarche-Gagnon G, Hawkings JR, Tedstone AJ, Wadham JL, Box JE, Beaton AD, Stibal M (2017) Carbon dating reveals a seasonal progression in the source of particulate organic carbon exported from the Greenland Ice Sheet. *Geophys Res Lett* 44:6209–6217

Lamarche-Gagnon G, Wadham JL, Sherwood Lollar B, Arndt S, Fietzek P, Beaton AD, Tedstone AJ, Telling J, Bagshaw EA, Hawkings JR, Kohler TJ, Zarsky JD, Mowlem MC, Anesio AM, Stibal M (2019) Greenland melt drives continuous export of methane from the ice-sheet bed. *Nature* 565:73–77

Lindbäck K, Pettersson R, Hubbard AL, Doyle SH, van As D, Mikkelsen AB, Fitzpatrick AA (2015) Subglacial water drainage, storage, and piracy beneath the Greenland ice sheet. *Geophys Res Lett* 42:7606–7614

Michaud AB, Dore JE, Achberger AM, Christner BC, Mitchell AC, Skidmore ML, Vick-Majors TJ, Priscu JC (2017) Microbial oxidation as a methane sink beneath the West Antarctic Ice Sheet. *Nat Geosci* 10:582-586

Musilova M, Tranter M, Wadham J, Telling J, Tedstone A, Anesio AM (2017) Microbially driven export of labile organic carbon from the Greenland ice sheet. *Nat Geosci* 10:360–365

Ruskeeniemi T, Engström J, Lehtimäki J, Vanhala H, Korhonen K, Kontula A, Liljedahl LC, Näslund J-O, Pettersson R (2018) Subglacial permafrost evidencing re-advance of the Greenland Ice Sheet over frozen ground. *Quat Sci Rev* 199:174-187

Stibal M, Wadham JL, Lis GP, Telling J, Pancost RD, Dubnick A, Sharp MJ, Lawson EC, Butler CEH, Hasan F, Tranter M, Anesio AM (2012) Methanogenic potential of Arctic and Antarctic subglacial environments with contrasting organic carbon sources. *Glob Change Biol* 18:3332–3345

van As D, Hasholt B, Ahlstrøm AP, Box JE, Cappelen J, Colgan W, Fausto RS, Mernild SH, Mikkelsen AB, Noël BPY, Petersen D, van den Broeke MR (2018) Reconstructing Greenland Ice Sheet meltwater discharge through the Watson River (1949–2017). *Arct Antarct Alp Res* 50:e1433799

van de Wal RSW, Russell AJ (1994) A comparison of energy balance calculations, measured ablation and meltwater runoff near Søndre Strømfjord, West Greenland. *Glob Planet Change* 9:29-38

Wadham JL, Hawkings JR, Tarasov L, Gregoire LJ, Spencer RGM, Gutjahr M, Ridgwell A, Kohfeld KE (2019) Ice sheets matter for the global carbon cycle. *Nat Commun* 10:3567

Whiticar MJ (1999) Carbon and hydrogen isotope systematics of bacterial formation and oxidation of methane. *Chem Geol* 161:291–314

---

## Author Comment (AC1) · 22 Sep 2020

GENERAL COMMENTS:

Pain et al. report carbon dioxide (CO2) and methane (CH4) concentrations, 13C stable isotopes and water chemistry from subglacial meltwaters of three glaciers located in southern regions of Greenland. The authors do a good job of describing these systems and discussing controls on both CO2 and CH4 dynamics in glacial systems. The manuscript is generally well written and easy to follow. In my opinion, this is a worthwhile contribution to the literature on the biogeochemistry of glacial systems. My suggested changes are mostly technical, but I distinguish below between "Specific Comments", referring to systematic changes in multiple sections of the manuscript or more substantive changes, from "Technical Comments", which are editorial.

> Thank you for your positive feedback regarding this study and its contribution to the understanding of subglacial biogeochemistry and carbon dynamics. We believe the suggestions provided by this review, namely to indicate with greater clarity the novel aspects of this study, the statistical tests conducted, as well as expand our discussion to include other possibilities besides a two-end member mixing model for $CO_2$ concentrations at our Isunnguata sampling site, will substantially improve the manuscript. We address specific and technical comments in detail below.

SPECIFIC COMMENTS: Could the novelty of the study be better highlighted? A significant body of work exists on the southwestern glaciers in particular (Innunguata, Russell), as described on lines 103-113, so it's not entirely clear to the reader what the novel contribution of this study is from the text at the outset.

> Yes, we will make efforts to highlight the novelty of this study, which is to assess the heterogeneity in greenhouse gas ($CO_2$ and $CH_4$) compositions of subglacial discharge of the Greenland Ice Sheet. Previous studies have evaluated $CO_2$ (Ryu and Jacobson, 2012) from the Isunnguata sub-catchment in this study, $CH_4$ concentrations in atmosphere near the Isunnguata sub-catchment (Christiansen and Jørgensen, 2018), $CH_4$ microbial cycling at the Russell (Dieser et al., 2014), and $CH_4$ concentrations at the Leverett Glacier (Lamarche-Gagnon et al., 2019). This is the first study to compare both $CO_2$ and $CH_4$ concentrations during the same time periods of Isunnguata and Russell glacier discharge, which demonstrates the regional heterogeneity in subglacial carbon dynamics in glaciers discharging into the Watson River, as well as heterogeneity between these southwest locations and southern Kiattut Sermiat site. We think it is important to demonstrate not only that this heterogeneity exists, but also that it represents a large range of greenhouse gas fluxes from subglacial systems that are controlled by various processes, including hydrologic and microbial processes as well as mineral weathering reactions. The significance of this finding is to point out the potential range of greenhouse gas fluxes in a warming world with retreating ice sheets, as is occurring today, as well as following the Last Glacial Maximum. These results have two important implications: they first provide

a range of potential impacts on atmospheric greenhouse gas compositions during ice sheet collapse after the Last Glacial Maximum. They additionally emphasize the need for caution in upscaling efforts of greenhouse gas fluxes from GrIS melt as polar amplification of global warming increases current rapid melting of the Greenland Ice Sheet.

Are errors throughout reported as standard deviations or standard errors (e.g., see section 3.1)?

Errors are all reported as standard deviations. This will be added to text in a revised version.

The samples were collected over a couple of years, which is perfectly reasonable for Arctic sites given the finances and logistics of working in the region. However, since the samples collected in 2018 were from the summer, and the 2017 samples were from the spring and fall, it would seem inappropriate to display the points with adjoining lines as a time series (i.e., Figures 3,4, 6, 7), because there can be large interannual differences in meltwater dynamics. There is no perfect solution to this, except to remove the lines adjoining the 2017 and 2018 samples, and perhaps discuss differences between the two years given the DMI climate data for both regions and/or the PROMICE discharges for the two southwest glaciers.

We agree that connecting the 2017 and 2018 data points may misrepresent our data set and will therefore redraw the figures to more clearly distinguish different sampling years. The suggestion of describing melting dynamics between 2017 and 2018 data using DMI and PROMICE discharge is excellent and will be very helpful in our discussion of temporal variations in gas concentrations throughout the melt season.

The justification for the measurement and presentation of the NH4+ data aren't obvious (only stated on L430). This should be explicitly indicated in the methods, but the data are not particularly informative and could be excluded (though this is entirely up to the authors).

We will provide justification for the inclusion of this parameter in the methods section. While glacial N cycling is complex (Wadham et al., 2016), $NH_4$ is produced during organic matter remineralization and may be used as a tracer for heterotrophic metabolism as there are a limited number of abiotic $NH_4$ sources. We use associations between $CO_{2-total}$ and $NH_4$ in Figure 8b as supporting evidence suggesting that organic matter remineralization is an important subglacial $CO_2$ source.

L271-275, Fig. 8: The statistics presented in the text vs. Fig. 8 are a bit confusing to follow. The text states that it's a correlation (i.e., independence of x and y), but a linear regression (i.e. dependence between x and y) is shown. Correlation statistics (i.e., r instead of r2) should be shown, or the text should be changed to reflect a linear regression (which is otherwise used throughout the text). Further, from Fig. 8a, the statistics appear to apply to the entire dataset as there is no indication from the caption or figure otherwise; however, the text states that the relationship was only observed for the Isunnguata samples, suggesting that the statisC2 tics presented only refer a subset of the data presented. I also wonder about the validity of removing the outlier... It's possible that this relationship is not linear, but rather parabolic, with lower 13C

values at lower CO2 concentrations indicative of rapid weathering, which can mimic the 13C-DIC signature associated with OM remineralization. There are not enough data to test this, but it would be something to keep in mind as the deviation of the "outlier" is not large enough to be indicative of analytical issues, but perhaps a true pattern. For this reason, I'm a little hesitant about the inference of a simple two end-member mixing model, especially since it does not seem to hold for the other sites.

> We will clarify the text regarding the statistics used. All statistics presented are linear regressions rather than correlations, and in all cases the regressions are site-specific and data from multiple sites are not combined.

> The possibility of a parabolic relationship between $\delta^{13}C\text{-}CO_2$ and $CO_{2\text{-total}}$ could explain our data and would imply that no data points are outliers. We will include discussion on other possibilities besides a 2-end member mixing model for Isunnguata and our other sampling locations. However, the otherwise strong relationship between $\delta^{13}C\text{-}CO_2$ and $CO_{2\text{-total}}$, as well as significant relationship between $CO_{2\text{-total}}$ and $NH_4$ for Isunnguata samples suggest that organic matter remineralization is the principal source of $CO_2$ in the subglacial environment. We acknowledge that the low $CO_2$ end member (or end members) in our mixing model is poorly constrained, and more data may be necessary to determine whether a 2-end member mixing model is valid.

TECHNICAL COMMENTS:

L26-27: Add reference to sentence starting with "Variations. . ." But also see Tranter et al. 2002, that has discounted a substantial role for glacial weathering on atmospheric CO2 concentrations over geological time scales (https://doi.org/10.1016/S0009- 2541(02)00109-2).

> Thank you for this suggestion, we will include this reference in a revised version.

L32-35: It might be useful to explain here the possible sources of CO2. As is, it seems somewhat disjointed from the preceding sentence, which describes CO2 budget. Both CO2 and CH4 will contribute to the carbon budgets of the system. The sources of CO2 are discussed in the following paragraph, but perhaps just a rejigging of this text would read more fluidly. One option would be to move L32-39 after the following paragraph and slightly expand upon the CH4 introduction before introducing the purpose of the paper. For example, of additional relevance to CH4 in subglacial environments is the formation of the necessary precursor H2 by rock comminution in Telling et al. 2015 (https://doi.org /10.1038/ngeo2533).

> Thank you for this helpful feedback, we will rework this section to improve the clarity of presentation.

Section 2.1: What is the seasonality of these systems? How much of the annual discharge occurs during the period where these were sampled? Is there winter flow? Glacial outburst floods?

> We will address seasonality of discharge in the revised manuscript as suggested in this review, which will provide information about discharge in 2017 and 2018 as well as

comparatively been these two melt years. Winter flow at these sites is low, with minor flow at the main outlet of the Isunngua River (Isortoq River), and no flow in the Watson River (Pitcher et al., 2020).

L103-113: See also Dubnick et al. (2017; https://doi.org/10.1002/2016JG003685), which includes calculated CO2 undersaturation at Kiattut Sermiat.

Thank you, we will include this reference for comparative purposes

L115: What were the specific sampling dates? If this is too much detail to have as text, then at least the number of sampling campaigns at each site each year would be useful information here.

We will include in a revised manuscript (or supplementary material) the number of samples per site per campaign and the date of each sample collection.

L240: What was the 13C-CO2 value for summer across all sites?

The $\delta^{13}C$-$CO_2$ values we collected at peak melt season (considered here to be during the month of July) were -14.1‰ and -12.1‰ for the Isunnguata, -12.4‰ for the Russell site, and -14.7‰ for the Kiattut Sermiat site.

Figure 1: Subscripts indicating carbonic or sulphuric acid (CA/SA) should be defined in the caption or on the figure.

This correction will be made.

Figure 2b: I'm wondering if there would be a way to trace the Watson River. It is difficult to see how the two study glaciers feed into the river.

We will provide an updated map figure in our revised version that will more clearly identify the sampling sites and names of tributaries and river segments that feed into the Watson River based on the feedback in this and another reviewer's comment.

Figure 2c: Kiattut Sermiat is spelled differently (Kiagtut) in panel c than elsewhere in the manuscript.

This correction will be made.

Figure 7: This caption should be more informative so that the figure can stand alone without the text. The y-axis is not intuitive without the definition for CO2total provided in the text (L266-267). As is, it's confusing because it looks like CO2 concentrations are negative, which in principle is impossible, though I understand what the figure shows.

Thank you for this feedback. This figure will be revised to more clearly indicate the meaning without relying on the caption text.

Figure 7: Could colour blocks instead of circle symbols be used for the legend? It's a small detail, but otherwise only technically refers to the Isunngata panel.

Yes, this revision will be made.

Figure 9: Symbology of the regression lines should be different from for the separation between years in which samples were collected.

We will change the format of the regression line to make this distinction.

**References**

Christiansen, J. R. and Jørgensen, C. J.: First observation of direct methane emission to the atmosphere from the subglacial domain of the Greenland Ice Sheet, Sci. Rep., 8(1), 2–7, doi:10.1038/s41598-018-35054-7, 2018.

Dieser, M., Broemsen, E. L. J. E., Cameron, K. A., King, G. M., Achberger, A., Choquette, K., Hagedorn, B., Sletten, R., Junge, K. and Christner, B. C.: Molecular and biogeochemical evidence for methane cycling beneath the western margin of the Greenland Ice Sheet, ISME J., 8(11), 2305–2316, doi:10.1038/ismej.2014.59, 2014.

Lamarche-Gagnon, G., Wadham, J. ., Sherwood Lollar, B., Arndt, S., Fietzek, P., Beaton, A. D., Tedstone, A. J., Telling, J., Bagshaw, E. A., Hawkings, J. R., Kohler, T. J., Zarsky, J. D., Mowlem, M. C., Anesio, A. M. and Stibal, M.: Greenland melt drives continuous export of methane from the ice-sheet bed, Nature, 565(7737), 73–77, doi:http://dx.doi.org/10.1038/s41586-018-0800-0, 2019.

Pitcher, L. H., Smith, L. C., Gleason, C. J., Miège, C., Ryan, J. C., Hagedorn, B., van As, D., Chu, W. and Forster, R. R.: Direct Observation of Winter Meltwater Drainage From the Greenland Ice Sheet, Geophys. Res. Lett., 47(9), 1–10, doi:10.1029/2019GL086521, 2020.

Ryu, J. S. and Jacobson, A. D.: CO2 evasion from the Greenland Ice Sheet: A new carbon-climate feedback, Chem. Geol., 320–321, 80–95, doi:10.1016/j.chemgeo.2012.05.024, 2012.

Wadham, J. L., Hawkings, J., Telling, J., Chandler, D., Alcock, J., O'Donnell, E., Kaur, P., Bagshaw, E., Tranter, M., Tedstone, A. and Nienow, P.: Sources, cycling and export of nitrogen on the Greenland Ice Sheet, Biogeosciences, 13(22), 6339–6352, doi:10.5194/bg-13-6339-2016, 2016.

---

## Author Comment (AC2) · 22 Sep 2020

Review of 'Heterogenous CO2 and CH4 content of glacial meltwater of the Greenland Ice Sheet and implications for subglacial carbon processes'

> We thank the reviewer for his comments and have addressed them in detail below. We see that the review describes three major issues. One is based on the hydrologic characteristics of our field sampling sites (and field site names as mentioned in other reviews). This issue is linked to a second issue related to atmospheric-stream water exchange between subglacial outlets and the stream water. The final issue is our presentation of effects of sub-glacial processes on production and consumption of methane and $CO_2$. We agree that many of these points are important to expand upon in our discussion while simultaneously recognizing and evaluating potential speculation. However, regardless of these unknown factors associated with the field sampling and locations, we think the data presented here still yield important information concerning the heterogeneity of greenhouse gas production and consumption caused by multiple sub-glacial processes, which we discuss, and that the heterogeneity complicates assessments of impacts to atmospheric compositions during glacial retreat, for example since the Last Glacial Maximum and into the future in a rapidly warming Arctic with continued ice sheet retreat.

The authors measured CO2 and CH4 concentrations and C stable isotopic signatures in meltwater samples from three marginal catchments of the Greenland ice sheet (GrIS) to evaluate sources and sinks of these important greenhouse gases under the fast melting ice sheet. The study is timely, wellstructured and -written, and uses adequate methods (with exceptions described below). However, interpretation of some of the results (especially the CH4 part) relies on incorrect assumptions and/or is unsupported by data, and is therefore too speculative at best. The issues listed below need to be carefully addressed before the study can be published.

First, the authors misunderstand and/or misrepresent the regional hydrology of the Kangerlussuaq area. The large outlet Isunnguata Sermia drains into the Isortoq River/Isortup Kuua (N of the Watson River catchment); its catchment is indeed quite large and extends deep into the ice sheet (probably to the ice divide). However, the water samples collected in this study did not originate from this large catchment and using the name Isunnguata Sermia is erroneous. Rather, the authors' IS site falls into the Point660 subcatchment sensu Lindbäck et al 2015 (as the authors admit at line 88), with an area of ca 30-60 km2 . This is part of the Russell Glacier catchment system (sometimes treated as part of the Leverett Glacier catchment, eg in Lindbäck et al. 2015), which is complicated, but likely does not reach far into the ice sheet. Calculating the Russell catchment area (for the RU site) as the difference between two individual and independent estimates of the Leverett catchment is incorrect (for example, some authors estimated the Leverett catchment at >1000 km2 ); the catchment feeding the Russell river is likely much smaller than 300 km2 (see eg van de Wal and Russell 1994). Figure 2b attributes the name Watson River to the river system originating at Point 660 and joining the larger river discharging from Leverett Glacier, which is problematic. Whereas the river terminology in the area is indeed somewhat confused, Watson River usually refers to the last short section of the river formed by the confluence of Akuliarusiarsuup Kuua and Qinnguata Kuussua which then enters Kangerlussuaq where the hydrology data used in the study (from van As et al 2018) were taken.

We agree these suggested changes are important to be consistent with previous geographic naming schemes. However, our assessments do not include flux calculations or specific yield estimates, which would depend on the catchment area of the glaciers studied, and proglacial processes are not in the scope of these manuscript. We will make the changes suggest as outlined below, but the changes in nomenclature will not greatly impact the presentation or interpretation of results or our findings, which show differences in subglacial carbon processes that result in heterogeneous concentrations of $CO_2$ and $CH_4$ in subglacial discharge.

The Isunnguata (IS) catchment we refer to is named as such because the watershed lies in between the Isunnguata and Russell (RU) glaciers, as described in delineations by Rennermalm et al. (2013) and the IS site is located at the southern edge of the Isunnguata Glacier. Previous work at the IS and RU sites in this study suggest differences in the subglacial lithology resulting from a geological contact in this area between the Nagssugtoqidian Mobile Belt (NMB) and Archaean Block (AB; Deuerling et al., 2019). These data indicate that our IS site predominantly drains a subglacial environment with a lithology more like the main Isunnguata Sermia catchment, which is believed to be mostly NMB, while our site at the Russell Glacier (as well as Leverett Glacier) drains mostly AB rocks (Fig. R1; Deuerling et al., 2019). We will include in revisions the uncertainties mentioned in the catchment size of the Russell but believe that the Isunnguata catchment in this study is appropriately named. We will also include a more thorough description of the tributaries to the Watson River, with an updated map to indicate the differently named segments of the river.

[Figure]

**Figure R1. Map indicating location of IS and RU sampling points (red stars) and the approximate lithologic boundary between the Nagssugtoqidian Mobile Belt (NMB) and Archaean Block (AB) (dotted red line). Figure modified from Dawes (2009) and Deuerling et al. (2019).**

Moreover, it is difficult to compare the study sites with previous works as the authors neither give details of their positions nor show any pictures. I assume the IS site is identical to the site used in Christiansen & Jørgensen (2018). The CH4 supersaturation referred to at lines 107-108 was measured in the air, with respect to atmospheric concentration, not in the water. The stream itself is very small (discharge at the portal was ca 1 m3 /s at peak melt in July 2018) and partly/mostly of supraglacial origin (a supraglacial/marginal stream can be found flowing into the ice margin several hundred meters above the portal). In terms of the RU site, it is not clear whether the authors sampled the main stem of the Russell Glacier river or one of the short subglacial tributaries flowing into it from the Russell Glacier margin (as depicted eg in Dieser et al 2014). The Russell Glacier river in this area has already flowed through a number of lakes and the CH4 signal would be difficult to interpret as purely subglacial. The order of magnitude difference in CH4 concentrations reported here and in Dieser et al (2014) is not discussed in the text but suggests that indeed samples were collected from the main stem of the river. A better description of the sampling sites is essential for an adequate assessment of the authors' interpretation.

All samples were collected from the respective main river stem as close as possible to the subglacial discharge site. Water did not flow through lakes directly in between glacial outlet sites and sampling sites at Isunnguata and Russell sites, though lakes are present further upstream of the Russell site and a broad and slow-flowing river channel was present between the glacial outlet and sampling point at Kiagtut Sermiat (referred to as a lake in this review). However, in all cases, dissolved $CO_2$ and/or $CH_4$ were out of equilibrium with respect to atmospheric concentrations, indicating that subglacial processes altered concentrations and that exchange with the atmosphere between the glacial outlet and sampling site did not erase (but we acknowledge may have altered) the subglacial carbon signal. While exchange would impact gas concentrations and isotopic compositions, there are multiple lines of evidence that suggest the observed chemical signals are predominantly controlled by subglacial processes, which we discuss further throughout this review (e.g. p. 9).

We agree that in addition to the GPS coordinates that were provided in the original manuscript, better descriptions, including the inclusion of pictures of sampling sites, would be valuable to maximize inter-comparisons between this and other nearby studies. We include pictures and more detailed site descriptions below.

[Figure]

**Figure R2. IS sampling site (photo taken July 15th, 2018)**

Isunnguata samples were collected from a site downstream of Point 660 where water flows from a subglacial discharge site that produced a visible boil at high discharge. No flow occurred further upslope of the boil, which was most clearly observed during the peak melt season (photo taken July 15th, 2018), suggesting it is the principal water source to the stream. As indicated by the photo, the water contains high concentrations of suspended sediment and is predominantly subglacial, with only minor observed contributions of supraglacial meltwater. This location is very close to the site discussed in Christiansen and Jørgensen (2018), where high atmospheric $CH_4$ concentrations were interpreted to reflect $CH_4$ supersaturation of the subglacial meltwater discharged at this site, which we now document in this manuscript.

[Figure]

**Figure R3. RU sampling site, photo taken on July 22nd, 2018**

Russell Glacier water samples were collected just downstream of the above pictured ice wall (Fig. R3) and thus water did not flow through any lakes directly in between the glacial outlet and the sampling location in this segment of the stream. While sampled water is a combination of recently discharged subglacial meltwater from under the Russell Glacier and proglacial discharge from further upstream (contributions from the Isunnguata at point 660 and other subglacial outlet sites as this segment of the river flows long the toe of the Russell glacier), gas concentrations are out of equilibrium with respect to atmospheric concentrations, suggesting the gas signal reflects subglacial processes because sites just upstream of this site are close to equilibrium with respect to atmospheric $CO_2$ and $CH_4$, with distinct increases in $CH_4$ concentrations downstream of the ice wall, indicating subglacial water contributions. While we did not include sites upstream of the RU glacial discharge site in this paper, gas concentrations at upstream locations can be found at doi:10.18739/A2PC2T94. If appropriate, we will present these data in a revised manuscript or supplemental information.

[Figure]

**Figure R4. KS sampling location (photo taken June 16th, 2017)**

The KS sampling location occurred as close as possible to the glacier outlet, seen in the center background in the picture above. The distance between the glacial outlet and our sampling site is roughly 1 km, however the river flow is slow with no rapids and therefore gas exchange should be minimal compared to that of the Watson River, in which rapids lead to more degassing. We additionally observe constant undersaturation of $CO_2$ with respect to the atmosphere at this site, indicating that water has not yet equilibrated with respect to atmospheric gas concentrations. Since $CO_2$ was still out of equilibrium with the atmosphere at our sampling point, $CH_4$ should also have been out of

equilibrium if it was considerably different from atmospheric concentrations in the subglacial environment.

While the inaccuracies in the description of the regional hydrology are easy to fix, they led the authors to a more problematic aspect of the study: correlating the CH4 and CO2 concentrations and isotopic signals from the IS and RU sites with discharge data from the Watson River. While both the Russell Glacier and Watson rivers show clear diurnal variations in discharge, large scale dynamics (including subglacial outbursts) observed in the large rivers is missing in the smaller Russell Glacier river, possibly due to the buffering effect of the lakes along the course of the river, as illustrated in the figure below comparing unpublished discharge data in m3 /s from LG (Leverett Glacier river), RU (Russell Glacier river), and IS (Isunnguata Sermia river/Isortup Kuua) in the summer 2018. The Watson River discharge dynamics may be further complicated by the unaccounted for Qinnguata Kuussua, which provides more than half of its water. This is in contrary to the authors' assumptions (line 217).

To my knowledge, this dataset has been made available to the authors, and it might be beneficial for the authors to use it for their correlations. Maybe it was not used because there were just 2 and 3 samples collected at RU and IS, respectively, in 2018 (as shown in Figures 3, 4, 6, 7, 9)?

The reason we do not use the dataset that is referred to in this review is because it has not published yet and would only be applicable for 2018. However, we agree that the varying scales of water contributions from the Isunnguata and Russell glaciers compared to that of the Watson River could result in non-proportional discharge between individual glacier discharge sites and the Watson River. These differences could impact the correlations we observe between gas concentrations and discharge. We address this issue by comparing Watson River daily discharge collected by PROMICE (As et al., 2018) to data collected by Rennermalm et al., (2013), which measured discharge just downstream of our Isunnguata sampling location (Fig. R5). We compare average daily discharge between both sites for the first year that complete melt season datasets were available from both sites (2009).

[Figure]

**Figure R5. Isunnguata (IS) site from this study, indicated with red star, compared to location of gauging station (AK4) in Rennermalm et al. (2013). We will use 2017 and 2018 data from this gauging station to address concentration-discharge relationships in the IS catchment. The catchment of this watershed lies between the**

**Isunnguata and Russell glaciers as indicated above and its area is estimated between 36-64 km² (figure modified from Rennermalm et al., 2013).**

[Figure]

**Figure R6. Comparison of discharge data between the Watson River (PROMICE) and the Isunnguata/Point 660 catchment (Rennermalm et al., 2013). Average daily discharge is compared between June 9 and September 1, 2009**

While the discharge values are significantly positively correlated (Fig. R6; p<0.0001), suggesting that assuming a rough proportional relationship between the smaller Isunnguata sub-catchment and the much larger Watson River is reasonable, the correlation is only of moderate strength. We therefore agree that concentration-discharge relationships would be better represented by discharge data from the Isunnguata sub-catchment. We are therefore collaborating with Asa Rennermalm, who will be a co-author on the revised manuscript, and who has collected discharge data from the Isunnguata sub-catchment over 2017 and 2018 that will soon be available. While we do not have such information for the Russell Glacier, Watson River discharge increases between the Isunnguata catchment and the Russell Glacier, making Watson River discharge a more appropriate approximation than for the Isunnguata. We test this assumption by comparing Russell discharge data (from unpublished source indicated in this review) from the record indicated with Watson River discharge in 2018. While only a few weeks of Russell glacier discharge is available, Russell average daily discharge is significantly positively correlated to Watson River average daily discharge (p<0.001), suggesting that assuming a proportional relationship is appropriate (Fig. R7). Additionally, because our analyses are not intended to be quantitative (e.g. no flux estimates are reported), and simply to address the relationships between gas concentrations and discharge, we believe that this approximation will still yield valuable insight into subglacial $CH_4$ and $CO_2$ dynamics in this region.

[Figure]

**Figure R7. Watson River compared to Russell Glacier average daily discharge between June 20 and July 15, 2018 (cite source), during which interval both records are available. Watson River discharge information is from (van As et al., 2018) and Russell Glacier discharge is unpublished (personal communication).**

The low number of samples and the fact they come from two different years, 2017 and 2018, is another weakness of the study. Figures 3, 4, 6, 7, 9 appear as though they show time series (suggested by the lines connecting the dots), which is not the case. I suggest the authors redraw the figure so it's clear the data come from two independent seasons. This should also be acknowledged in the text, and the possible differences in hydrology and the potential impact on the export of gases discussed more in detail. For example, Hawkings et al (2015) showed large interannual variations in total discharge and solute and particulate fluxes from the Leverett catchment between 2009-2012. Also, no statistical analysis of the results was done and it's unclear whether the differences in water chemistry were significant between the streams – was this due to the low number of samples?

Because our intention was to capture changes in water chemistry between disparate locations and throughout the melt season, with a primary goal of identifying differences between glacial discharge sites, logistical constraints did not allow us to sample any individual subglacial discharge site throughout the melt season. While not included in our original manuscript, we conducted statistical analysis through one-way ANOVA, which indicated that both $CH_4$ and $CO_2$ concentrations differ between sites (p<0.0001), supporting the principal findings stated in this manuscript. Dissolved gas isotopic compositions, however, do not significantly differ. We will include these statistical tests in the results and discussion of a revised manuscript. We will additionally redraw the figures to clarify that samples were collected over multiple years, which also addresses a comment by Reviewer 1.

This should also be acknowledged and/or explained. Analysis of stable isotopic signatures of C in CH4 and CO2 is a powerful tool for determining the origin of the respective gases; however, caution must be exercised when interpreting the results for glacial meltwater samples. Glacial

meltwater is a very dynamic mixture consisting of several components with different origins: the water mostly comes from the surface and so has been in direct contact with the atmosphere (and its CO2); the sediment is predominantly of subglacial origin. Dissolved CH4 also likely originates from the subglacial environment, while CO2 has multiple sources -- as the authors show and discuss throughout the ms. The isotopic separation factor $\varepsilon C$ (ie $\delta$ 13CCO2 - $\delta$ 13CCH4), used in this study to assess the sources and sinks of CH4, is suitable for closed systems (as defined in Whiticar 1999), but caution must be exercised when using it for glacial meltwater and the limitations should be acknowledged and discussed in the text. The authors also calculate the fraction of oxidised methane (fox) using a number of assumptions, some of which might not be substantiated. For example, Michaud et al (2017) modelled the kinetic isotopic fractionation factor $\alpha ox$ beneath the West Antarctic Ice Sheet, an environment likely to be more representative of the bed of the GrIS, at 1.004. The authors use a value of 1.049, which may lead to an underestimation of microbial oxidation of CH4 in the GrIS subglacial system. More importantly, outgassing, as a major process affecting meltwater gas concentrations, should not be ignored. In the turbulent flow of glacial rivers, most CH4 will likely outgas very quickly: for example, in the Leverett Glacier river, virtually all CH4 is gone after ca 2 km (Lamarche-Gagnon et al 2019). Moreover, outgassing affects not only the concentrations, but also the isotopic composition of gases due to fractionation (see eg Banks et al 2017), driving the remaining dissolved gas to more positive (heavier) values. This may result in an overestimation of CH4 oxidation. While outgassing was probably less significant at IS (as the authors sampled only 10 m from the subglacial outlet), it may have affected gas concentrations at RU (100 m) and definitely would have at KS (>1 km and a proglacial lake; see below). Much more attention should be paid to the possible effects of this process in the discussion. Moreover, it should be pointed out that outgassing is likely enhanced in glacial systems by considering the large pressure differentials between the subglacial environment where the CH4 is produced and the atmosphere, and the rapid depressurisation that results from pressurised subglacial waters exiting the ice. Such depressurisation effect is likely to also influence the isotopic signature of the sampled gases in runoff (Banks et al 2017). While accounting for outgassing/depressurisation and their effects on isotopic fractionation might be difficult, if not impossible, the authors should at least discuss the limitations and biases of not doing so, and whether or not the assumptions from their oxidation model would still hold true.

> We agree that outgassing could occur between the subglacial outlet portals and our sampling locations. However, our discussion of subglacial methanogenesis sites focuses on the Isunnguata and Russell sites, where CH4 supersaturation was observed, and from which samples were collected in close proximity to the glacial discharge site. While some outgassing certainly occurs during water transit from the subglacial environment to the sampling location, the extent of outgassing should be relatively constant over the melt season, and thus would not explain temporal trends in fox or $\varepsilon_c$. Additionally, if fox and $\varepsilon_c$ were predominantly reflective of outgassing processes, the isotopic compositions of CH4 from the Russell would lead to consistently higher estimates and fox and lower $\varepsilon_c$, however both variables are nearly identical between Russell and Isunnguata sites (manuscript Fig. 4), suggesting that isotopic compositions reflect similar subglacial processes between the two proximal sites. Nevertheless, the reviewer makes an important point here, which we agree with, and we will include additional discussion of how

possible outgassing may affect our interpretations and highlight the expected impacts on isotopic compositions.

Our calculations of $CH_4$ oxidation were intended to provide a minimum estimate of the amount of subglacial $CH_4$ oxidation that would be implicated by the observed $CH_4$ isotopic compositions. To do this, we used the upper limit of $CH_4$ oxidation fractionation factors in the literature. Since we do not have enough information to constrain the actual fractionation factor, we believe that presenting a minimum value, even if it is likely to be an underestimate, provides a valuable constraint on the inferred role of subglacial $CH_4$ oxidation. While other fractionation factors, such as that presented in the cited Antarctic study, may be closer to the value of the true fractionation factor under the Greenland Ice Sheet, using this value introduces more assumptions than our current approach. Assuming similar subglacial environments between Antarctica and Greenland may additionally not be appropriate because subglacial conditions differ between these settings. For instance, while supraglacial meltwater flows from the surface to the base of the Greenland ice sheet and provides biologically relevant material such as labile organic matter, oxygen, and nutrients in dust debris or imparted through atmospheric deposition, no such supraglacial meltwater transfer has been observed in Antarctic settings and surface water production in Greenland is currently much higher than in Antarctica (Bell et al., 2018). While subglacial liquid water may be present, in Antarctica, the glacial history is significantly different from Greenland and the time since atmospheric contact in Antarctic subglacial lakes can be thousands to millions of years.

Despite the uncertainty in $CH_4$ oxidation fractionation factors, revisions could include a range of $CH_4$ oxidation estimates using the range spanned in the literature, if appropriate.

The KS site is additionally problematic as there is a large proglacial lake right by the portal, with an estimated water residence time in the order of 24 hours at peak discharge (Hatton et al 2019). This may significantly change the concentrations and isotopic signatures of the dissolved gases exported further downstream via outgassing (and possibly also microbial processes in the lake bottom sediment), and may be the reason why the CH4 concentrations at KS are near atmospheric equilibrium. This should also be mentioned when discussing the results from KS.

We agree with this assessment. However, flow at KS through the glacial lake is slow with very little turbulence, which should minimize the extent of outgassing. Additionally, since $CO_2$ is out of equilibrium with respect to atmospheric concentrations, we believe that the lack of $CH_4$ supersaturation is not due to outgassing. However, because outgassing is a possible contributing factor, we will discuss this in the text more clearly.

In the discussion, the authors interpret the observed orders of magnitude variability in dissolved gas concentrations in the meltwater samples as differences in the sources and sinks of the gases (lines 290-291). However, some of the explanations of CH4 variability are unnecessarily speculative and unsupported by data, and some rest on incorrect assumptions. First, the variability in subglacial OM substrates is invoked (312). This is certainly a factor affecting subglacial C cycling and export rates, but no supporting OC data are presented. Permafrost reservoirs, suggested based on the study by Ruskeeniemi et al (2018), are unlikely to be of

importance (and were not alluded to in LamarcheGagnon et al 2019, as suggested at line 314), as they extend only a few km into the ice sheet bed. Moreover, Ruskeeniemi et al (2018) only focused on the thermal state of the sediments/soils, rather than the nature of OC. I agree the Holocene ice margin fluctuations were probably important in providing fresh OC substrate that could have been metabolised into CH4 that is currently being exported. Older (Eemian) OC sources are however also present and exported in the meltwater (Kohler et al 2017) and may have been used as methanogenesis substrates.

> We do not include DOC or POC data in this manuscript because delineating OC sources (as subglacial versus supraglacial, for instance), which may have very different concentrations and reactivities between sites and over time, is outside the scope of this manuscript and data set. We do not believe that DOC data would substantially impact the results presented in this manuscript, as many other indicators of OC remineralization ($CO_2$, $CH_4$, $\delta^{13}C$-$CO_2$, and $NH_4$) all indicate that OC remineralization occurs under the ice sheet but to varying extents, and DOC concentration information is not needed to make this inference.

> We will modify the text regarding the formation and nature of the subglacial OC deposits that may be implicated by our results. We acknowledge that we have no data regarding these deposits, which are as of now poorly constrained in both their nature and distribution. However, the lack of information regarding subglacial OC deposits does not impact the interpretation of our results, which all indicate that varying OC remineralization (and mineral weathering) occurs and contributes to heterogeneous greenhouse gas content of subglacial discharge.

Reservoirs of old CH4 are not considered in the study. Second, a direct effect of oxygen supply to the ice sheet bed by surface meltwater on methane production/oxidation is proposed, based on the negative correlation of CH4 concentrations at RU and Watson River discharge (319-327). As explained above, linking gas concentrations and isotopic signatures at IS and RU to discharge data from the Watson River is misleading. In addition, the authors only consider live methanogenesis and ignore potential old CH4 storage/leakage (see above). Dilution by meltwater is only acknowledged at lines 328-329 as an alternative explanation, although it plays a significant role. The local subglacial sources of CH4 are probably limited to microbial activity (Lamarche-Gagnon et al 2019), which takes place in anoxic sediments buried under the ice. Whether it's recent activity or reservoirs of ancient CH4, its export is dependent on meltwater tapping and flushing pockets of produced gas. As a result, CH4 concentrations in the meltwater are necessarily discharge-dependent. This is indeed complicated by outburst events; however, these are limited to large outlets (lakes form at much higher altitudes further into the ice sheet than those to which this subcatchment extends), and explaining the lack of discharge-CH4 concentration relationship at IS by outbursts (330-332) is therefore is not justified. Last, CH4 oxidation, discussed at lines 350-361, is certainly an important process controlling the amount of CH4 that will be exported from under the ice to the atmosphere. However, in addition to the uncertainty in determining the degree of CH4 oxidation, the authors' interpretation of the data again relies on correlating the CH4 concentrations at IS and RU with Watson River discharge and on treating the 2017 and 2018 data as a time series, both of which are flawed (see above).

We acknowledge that our interpretations assume that most $CH_4$ is produced actively, though leakage of "old" $CH_4$ may occur, and will therefore include this as a possibility. However, particularly in the peak melt season when subglacial residence time should be shortest, our measured $\delta^{13}C$-$CH_4$ values closely match those measured by live methanogenic communities by Dieser et al. (2014), supporting (but not confirming) active methanogenesis under these portions of the Greenland Ice Sheet.

We disagree that dilution by meltwater is not presented as a key mechanism explaining the discharge-concentration relationship of $CH_4$ as it is one of only two mechanisms presented for the Russell Glacier that are given approximately equivalent amounts of discussion (Lines 325-327: higher methanogenesis during low flow resulting from greater subglacial residence time. Lines 327-328: dilution of methanogenic subglacial drainage by supraglacial meltwater). These mechanisms are additionally not mutually exclusive. Our modified manuscript using discharge data from the Isunnguata sub-catchment (called Isunnguata in this study) will likely somewhat change the discharge-concentration relationship at the Isunnguata. In our revised manuscript, we will more clearly describe the lack of relationships between discharge and concentration that invoke heterogeneous $CH_4$ distributions in distributed portions of the subglacial drainage network, which we will distinguish more clearly from potential impacts from outburst events.

In summary, I recommend the authors revisit their local hydrology description and interpretation, rename their sampling sites accordingly, avoid correlating their small stream data with the Watson River discharge record, and properly acknowledge the limitations and uncertainties of the used geochemical calculations for interpretation of the subglacial gas sinks and sources, especially for CH4

This review has provided valuable feedback regarding information needed for better clarity of our sampling sites and their hydrological settings, which we will heavily incorporate in manuscript revisions. Many of the criticisms regarding hydrology and use of Watson River discharge to assess concentration-discharge relationships will be addressed by new discharge information for our study site (Isunnguata sub-catchment, referred to as Isunnguata in this study) provided by Asa Rennermalm. Our interpretations may shift somewhat following these modifications although we think based on the small watershed correlation with Watson River discharge, our primary findings will be robust. Specifically, $CO_2$ and $CH_4$ concentrations exhibit a high degree of heterogeneity between glacial discharge sites of the Greenland Ice Sheet, and that variations likely occur due to disparate levels of subglacial organic carbon remineralization. We think it is important to demonstrate not only that this heterogeneity exists, but also that it represents a large range of greenhouse gas fluxes from subglacial systems that are controlled by various processes, including weathering reactions and hydrologic and microbial processes. The significance of this finding is to point out the potential range of greenhouse gas fluxes in a warming world with retreating ice sheets, such is occurring now, as well as occurred since the Last Glacial Maximum. These results could significantly impact upscaling efforts of greenhouse gas fluxes from GrIS melt, which will be an increasingly import carbon flux in the coming decades.

**Specific comments**

53: please specify if Graly et al 2017a or b

> This refers to Graly 2017b and will be modified in the revised manuscript.

58-60: relevant work should be cited here, eg the recent review by Wadham et al (2019)

> This citation will be added.

66: Musilova et al (2017) did not study subglacial microbial activity; this reference is irrelevant here

> This will be removed.

107-110: methanogens have also been identified in Russell Glacier basal ice (Stibal et al 2012) and Leverett Glacier river suspended sediment (Lamarche-Gagnon 2019); $CH_4$ supersaturation in meltwater was also measured by Dieser et al (2014) but not by Christiansen & Jørgensen (2019)

> Thank you for these corrections, we will revise our statements to reflect this.

306: Lamarche-Gagnon et al (2019) measured higher CH4 concentrations than 600 nM (up to 4000 nM during early season)

We used the maximum value as indicated in the continuous monitoring record and supplemental information (below). If a maximum of 4000 nM has been published elsewhere, we will modify our statement.

[Figure]

**Figures taken from Lamarche-Gagnon et al., (2019) indicating CH4 concentrations at the Leverett Glacier over the 2018 melt season (left) and comparison of sensor and manually measured CH4 concentrations (above).**

405: how do the results compare to the recent paper by Andrews et al (2018) focused on dissolved C dynamics in Russell Glacier meltwater, including the sources of subglacial CO2?

Andrews et al. (2018) estimated up to 30% subglacial DIC production by microbial activity in discharge of the Russell glacier, consistent with our study indicating significant contributions of subglacial organic matter remineralization. We will include discussion of these similarities in our revised manuscript.

446: please explain 'chemostatic behavior'

Chemostatic behavior indicates that concentration is constant over a range of discharge and indicates that concentrations are not controlled by dilution and are not transport-limited.

695: Figure 1 is a weird combination of 2D and 3D which makes it difficult to interpret. Also, could the authors provide references for CO2 and CH4 evasion through crevasses?

The exchange through crevasses is drawn to indicate the semi-closed nature of the subglacial system, whereby gas exchange may occur. While no studies have directly measured this exchange to our knowledge, atmospheric gas exchange in the subglacial environment has been implicated (Graly et al., 2017) and contact between the atmosphere and the subglacial environment invokes exchange in moulins or fractures that allow exchange of atmospheric gases to depth.

700: Figure 2 needs redrawing to correct the river network names and to better indicate the sampling sites; please also use the newer transcription 'Kiattut', to be consistent with the text.

This correction will be made.

740: the regression line in Figure 8b doesn't look right – were some points omitted?

No points were omitted, however there is one point in the $CO_{2\text{-atm}}$ series that falls below the other points but is partially obscured by a point in $CO_{2\text{-OM}}$, which may have resulted in confusion. The regression includes all points in the data series.

**References**

Andrews, M. G., Jacobson, A. D., Osburn, M. R. and Flynn, T. M.: Dissolved carbon dynamics in meltwaters from the Russell Glacier, Greenland Ice Sheet, J. Geophys. Res. Biogeosciences, doi:10.1029/2018JG004458, 2018.

As, D. Van, Hasholt, B., Ahlstrøm, A. P., Box, J. E., Cappelen, J., Colgan, W., Fausto, R. S., Mernild, S. H., Bech, A., Noël, B. P. Y., Petersen, D., Broeke, M. R. Van Den, As, D. Van, Hasholt, B., Ahlstrøm, A. P., Box, J. E., Colgan, W., Fausto, R. S., Mernild, S. H. and Mikkelsen, A. B.: Reconstructing Greenland Ice Sheet meltwater discharge through the Watson River ( 1949 – 2017 ) Reconstructing Greenland Ice Sheet meltwater discharge through the Watson, Arctic, Antarct. Alp. Res., 50(1), doi:10.1080/15230430.2018.1433799, 2018.

Bell, R. E., Banwell, A. F., Trusel, L. D. and Kingslake, J.: Antarctic surface hydrology and impacts on ice-sheet mass balance, Nat. Clim. Chang., 8, 1044–1052, 2018.

Christiansen, J. R. and Jørgensen, C. J.: First observation of direct methane emission to the atmosphere from the subglacial domain of the Greenland Ice Sheet, Sci. Rep., 8(1), 2–7, doi:10.1038/s41598-018-35054-7, 2018.

Dawes, P. R.: The bedrock geology under the Inland Ice: The next major challenge for Greenland mapping, Geol. Surv. Denmark Greenl. Bull., (17), 57–60, doi:10.34194/geusb.v17.5014, 2009.

Deuerling, K. M., Martin, J. B., Martin, E. E., Abermann, J., Myreng, S. M., Petersen, D. and Rennermalm, A. K.: Chemical weathering across the western foreland of the Greenland Ice Sheet, Geochim. Cosmochim. Acta, 245(245), 426–440, doi:10.1016/j.gca.2018.11.025, 2019.

Graly, J. A., Drever, J. I. and Humphrey, N. F.: Calculating the balance between atmospheric CO2 drawdown and organic carbon oxidation in subglacial hydrochemical systems, Global Biogeochem. Cycles, 31(4), 709–727, doi:10.1002/2016GB005425, 2017.

Lamarche-Gagnon, G., Wadham, J. ., Sherwood Lollar, B., Arndt, S., Fietzek, P., Beaton, A. D., Tedstone, A. J., Telling, J., Bagshaw, E. A., Hawkings, J. R., Kohler, T. J., Zarsky, J. D., Mowlem, M. C., Anesio, A. M. and Stibal, M.: Greenland melt drives continuous export of methane from the ice-sheet bed, Nature, 565(7737), 73–77, doi:http://dx.doi.org/10.1038/s41586-018-0800-0, 2019.

Rennermalm, A. K., Smith, L. C., Chu, V. W., Box, J. E., Forster, R. R., Broeke, M. R. Van Den and As, D. Van: Evidence of meltwater retention within the Greenland ice sheet, Cryosph., 1433–1445, doi:10.5194/tc-7-1433-2013, 2013.

---

## Author Comment (AC3) · 22 Sep 2020

Thank you very much for this feedback. We will revise our place names as suggested and will provide site pictures and more thorough descriptions to ensure our Isunnguata Sermia (IS) sampling site is sufficiently differentiated from the glacier's terminal outlet. Pictures, watershed delineations, and more thorough descriptions are currently given in our response to reviewers (Reviewer 2).

---

## Author Response (AR1)

Dear Dr Pain and co-authors,

Thank you for your response to reviews. I must apologise for the delay in responding to your comments. I have reviewed the responses and would now like to invite you to proceed with the upload of your revised manuscript.

Please consider the reviewers' comments carefully. In particular, Reviewer 2 expresses some concerns about the validity of your assumptions when interpreting the data. I, and the two reviewers, believe that you have an excellent and hard-won dataset, but both reviewers express some concerns in the interpretation. Please consider their advice carefully before resubmission. When you address the comments from the reviewers, to which you have detailed your responses, I also request the following amendments to the manuscript:

Thank you very much for your evaluation and helpful suggestions. We likewise apologize for the delay in uploading our revised manuscript. The revision included the inclusion of new hydrologic data that had to undergo quality control prior to inclusion in the manuscript, which extended the time needed for the revision.

We have taken many of the reviewer suggestions into consideration and believe our revised manuscript is significantly improved in the clarity in the presentation results and reduction of assumptions. The major changes to the manuscript include:

1) As suggested, we have renamed our "Isunnguata" site as "sub-Isunnguata" to reflect that this site is not the main drainage channel of the Isunnguata. The modification to the text can be found in lines 86-99.

2) Our previous version used Watson River average daily discharge to assess the relationships between discharge and greenhouse gas dynamics, however Reviewer 2 raised questions about the use of Watson River discharge data, which drains a large watershed area (up to 12,600 km$^2$; Lindbäck and others, 2015), compared to the individual drainage areas of the sub-Isunnguata (watershed area ~40 km$^2$; Rennermalm and others, 2013) and Russell Glacier (watershed area <900 km$^2$; (Hawkings et al., 2016). This new revision therefore uses discharge data from a segment of proglacial river downstream of the sub-Isunnguata sampling site (site AK4; Fig. 2b). Discharge records from this site have been used to evaluate the glacial hydrology of the sub-Isunnguata watershed in other studies (e.g. Rennermalm and others, 2013).

We also use AK4 discharge data to assess concentration-discharge relationships from the Russell Glacier. Although this site is upstream of the Russell Glacier, it is much closer to the Russell Glacier than the downstream Watson River PROMICE gauging station (van As et al., 2018) that receives meltwater contributions the Akuliarusiarsuup Kuua (draining the sub-Isunnguata, Russell, and Leverett catchments) as well as the much larger Qinnguata Kuussua. We therefore believe that Watson River discharge records are likely to be less representative of the temporal changes in the magnitude and variability of discharge from the much smaller Russell

glacier catchment. Since our evaluations are not intended to be quantitative, we believe using AK4 discharge records to evaluate Russell discharge-concentration relationships can still provide valuable information regarding the hydrologic controls of $CO_2$ and $CH_4$ dynamics in this region. This addition of sub-Isunnguata discharge data and the resulting interpretations represent significant improvements to our ability to assess hydrologic controls of greenhouse gas dynamics under the Greenland Ice Sheet, therefore we have made Asa Rennermalm a co-author on this manuscript.

3) We have presented the data from multiple years (2017 and 2018) as individual data series by removing lines between data points in plots that show chemistry versus day of year (e.g. Figs 3, 4, 6, 7). This change more clearly presents data as being collected over multiple years and addresses comments from Reviewers 1 and 2.

4) We have removed the results of a $CO_2$ isotopic mixing model (previously Fig. 9) because of the number of assumptions embedded in this analysis. The newly included discharge data for the sub-Isunnguata indicated greater variability in the relationship between discharge and concentrations of putative $CO_2$ sources, therefore we deemed this analysis too speculative to include in the revised version.

5) We have added a new figure to present discharge versus $CO_2$ concentration and isotopic compositions (Fig. 8). Concentration-discharge analyses for this dataset are now much more robust due to the new discharge information for the sub-Isunnguata watershed.

As well as considering the place name nomenclature suggested by Dr Graly and the nominated reviewers, please consider whether you can replace the 'Watson River' with the Greenlandic name, Qinnguata Kuussua or Akuliarusiarsuup Kuua, as appropriate. Also note that in Figure R5 in your reponses, you label Isunnguata Sermia Glacier: this is nonsensical (Sermia broadly translates as glacier), so please remove the extra 'Glacer'. I strongly support Reviewer 2's request to amend the naming of the 'Isunnguata Sermia', since your data do not represent the wider Isunnguata Sermia catchment. Perhaps you could consider subIsunnguata, Isunnguata-Russell or lateral-Isunnguata, or some other better version that does make it clear that this is a sub-catchment, not the main trunk which drains into Isortoq. The map in R5 is very useful for context, and perhaps an expanded version also showing the main trunk of Isunnguata Sermia to the north could be added to the map figure.

We thank you and the reviewers very much for indicating these important place name clarifications. We have modified the map figure (Fig. 2) to indicate the positions of the Isortoq, Qinnguata Kuussua and Akuliarusiarsuup Kuua, as suggested, and have preferentially used these names (rather than Watson River) throughout the text.

We have additionally renamed our Isunnguata sampling location as the sub-Isunnguata location. The site descriptions also now provide more detail about local hydrology (lines 86-105).

The markers on the map (Figure 2) are very large, meaning it is quite hard to see exactly where the samples were collected. Please could you reduce the size? I also find Figure R1 quite garish: the markers again overwhelm the map. The boundary could be more subtly delineated without detriment; although I do wonder at the utility of this map – I do not think the geological boundary exerts much influence over your sampling sites, since they both lie to the north.

> We have redrawn the map figure to address this issue (Fig. 2)

I agree with Reviewer 1 on the presentation of the NH4 data: it doesn't add much to the story, so wonder if it really needs to be in the manuscript. I appreciate your justification in the response, but I am unconvinced that the data really help in the exploration of the carbonate processes – it seems a rather weak association with the organic matter remineralisation hypothesis.

> We have taken this suggestion and removed the NH$_4$ data from the manuscript.

Does the Pitcher et al. 2020 paper not show that there is only winter flow in the main Isortoq outlet? From my understanding of your sample sites, and the clarifications in your responses, your samples were not collected from Isortoq, but from Akuliarusiarsuup, which experienced no winter flow. Please consider this in your revised description of the flow regimes (as requested by Reviewer 1).

> We have revised our description of the flow regimes in the region (lines 86-105) as well as the presentation of locations in the map figure (Fig. 2)

In your response to Reviewer 2, you include some site photos. These are very useful and I hope these will be included in the Supplementary info, or incorporated into the manuscript somehow. Please could you indicate the direction you are facing when the photo is taken? The Isunnguata one is hard to orient. One note: a 'boil' is not a common term in glacier hydrochemistry. Could this be a small subglacial upwelling (eg. Wadham et al. 1998; Irvine-Fynn and Hodson, 2010)? I also agree with Reviewer 2 in the assessment of the slow flowing section of the terminus of Kiatuut Sermia as a 'lake' rather than a slow flowing section of river (I have in fact kayaked and depth sounded this lake, see Beaton et al. 2017 and Bagshaw et al. 2014). It is hard to see from Figure 2 exactly where your sampling site was, but if it was downstream of this feature, it will impact the residence and transit time of water from the subglacial environment to the sampling site. The presence of this lake should be acknowledged in your revised manuscript.

> We include site photos in supplementary information, along with an indication of the direction the photo was taken. We also rename the boil as a subglacial upwelling, and revise our description of the lake at the Kiattut Sermiat site (lines 127-131) and in the supplementary information.

Your defence of the manuscript in the final paragraph of your response to Reviewer 2 is commendable. I wonder if some of this material could be used in your concluding statement in the main text? The current conclusions are very focussed on weathering implications, and do not really demonstrate the utility of this study to an audience beyond the glacier weathering

community, whereas your defence articulates it very well! Thank you for your patience with the review process; I look forward to reading the next version of the manuscript.

> Thank you very much for this suggestion. We have revised the conclusions to be more inclusive of the broader implications of this work beyond the mineral weathering community (lines 514-536). We very much appreciate your helpful feedback throughout the review process and believe the revised manuscript is a strong improvement from the previous submission.

Dr Liz Bagshaw Editor

---

## Author Response (AR2)

**Response to reviewers: "Heterogeneous $CO_2$ and $CH_4$ content of glacial meltwater from the Greenland Ice Sheet and implications for subglacial carbon processes" (Pain et al.)**

**Editor comments**

L11-14: an extremely long sentence featuring too many 'and's. Please divide the sentence for easier digestion.

> Agreed-- we have changed the sentence to: "We evaluate subglacial discharge from the Greenland Ice Sheet for carbon dioxide ($CO_2$) and methane ($CH_4$) concentrations and $\delta^{13}C$ values in order to evaluate subglacial $CH_4$ and $CO_2$ sources and sinks using geochemical models. We compare discharge from southwest (a sub-catchment of the Isunnguata Glacier, sub-Isunnguata, and the Russell Glacier) and southern Greenland (Kiattut Sermiat)." (lines 11-14)

L14: erroneous semicolon – should be a colon, or just a comma.

> Corrected

L16: 'meltwater in southwest sites' – should this be 'from' or 'at'?

> Changed to "from"

L64: not sure this sentence is necessary: the majority of our readers know that there are no other ice sheets remaining in the northern hemisphere. If you choose to retain, either Northern should not be capitalised, or both Northern and Hemisphere should be capitalised (I think the latter?).

> We opted to keep this in the manuscript as it reinforces the point that the current landscape of Greenland can be used to understand larger deglaciation events such as that which occurred since the LGM. We have changed to Northern Hemisphere

L91: typo

> Corrected

L119: spacing error prior to ref

> Corrected

P12: two paragraphs beginning with 'While', and multiple sentences beginning the same – suggest rewording one or two for readability.

> Wording has been changed in two instances (lines 346 and 365)

L500: typo - missing space

Corrected

**Reviewer 2 comments**

Title – correct to 'heterogeneous'

Corrected

Abstract – verb still missing in sentence at lines 20-22

Corrected

Intro
57 and elsewhere please specify if Graly et al 2017a or b

Graly 2017a was incorrectly included in the reference list, so now there is only one Graly 2017 reference.

62-64 relevant work should be cited here, eg the recent review by Wadham et al (2019)
70 Musilova et al (2017) did not study subglacial microbial activity; this reference is irrelevant here

Citations have been added

Methods
91 typo in 'the Qinnguata Kuussua'

Corrected

97 mentioning the Isunnguata catchment size is irrelevant and potentially misleading here and should be removed

We feel that it is important to contextualize the regional hydrology and describe the glaciers provided in the map, therefore have opted to keep this number. The use of the term sub-Isunnguata throughout the manuscript reinforces that this is a sub-catchment of the Isunnguata, and the difference between the catchment areas is described in lines 96-97.

102 The estimate of the Russell Glacier catchment size of 300 km2 has remained in the text. This is questionable and I strongly advise the authors to remove it or to add this is probably exaggerated (van de Wal and Russell 1994).

We have added language to communicate this uncertainty in lines 101-102 and added this citation.

Results

277-278 The correlations between Sub-IS discharge and δ13CCH4, εC, and fox are weak and the data points certainly do not suggest linear relationships (Fig 5bcd). The relevant discussion should be toned down accordingly (see below).

> We now describe the correlations as weak (lines 277-278)

Discussion
337-339 This explanation is confused. Methane production requires a very negative redox potential and any external terminal electron acceptors brought in by meltwater (O2, NO3-) would inhibit it and lead to CH4 oxidation instead. The EAs for methanogenesis are either CO2 (for the hydrogenotrophic pathway) or acetate (acetoclastic methanogenesis is a disproportionation). Please remove or rephrase this.

> Thank you for pointing this out-- this was an important error in the text. We have modified the text to reflect this: "If limited by residence time, a hydrologic link between glacial hydrology and subglacial biogeochemistry would be established because supraglacial discharge delivers terminal electron acceptors to the ice bed and would limit methanogenesis." (lines 337-339)

354-356 If CH4 is stored under the ice sheet no fractionation is likely to occur. Therefore, it is impossible to decide whether the released CH4 comes from active methanogenesis or old reservoirs, based only on δ13CCH4. Please remove or rephrase this.

> We have rephrased as follows: "The similar isotopic ratio between our samples and that measured in active methanogenic communities could indicate that similar methanogenesis pathways occur across this region, or that the $\delta^{13}$C-CH$_4$ of stored subglacial CH$_4$ has not been fractionated by oxidation or transport in the peak melt season when we observe these depleted $\delta^{13}$C-CH$_4$ values." (lines 352-354)

367 While I agree it is unlikely that the extent of outgassing would vary significantly between sampling times, it may be affected by discharge due to changes in turbulent flow. Please change to 'outgassing would not fully explain temporal differences'.

> This change has been made.

372-374 This paragraph is based on the weak correlations shown in Fig 5 (see above) and should be toned down.

> We now acknowledge the weak correlation here (line 373)

392, 415 please remove the sermiat/glacier pleonasms

> This change has been made.

423-8 unfinished sentences, please rephrase

Typos have been corrected.

491-498 More recent and appropriate literature should be referred to. For example, microbial sulphide and thiosulphate oxidation in the subglacial environment have been quantified by Boyd et al 2014 and Harrold et al 2016, respectively; the presence of growth substrates as a factor for CH4 production in subglacial samples has been shown experimentally by Stibal et al 2012; Wadham et al 2010 provided a thorough overview of subglacial weathering and the role of microbial processes in it.

These references are now included (lines 489, 491).

---

## Editor Decision (ED2)

Pain et al. Editor's recommendations

Dear Dr Pain and colleagues,

Thank you for your considerable efforts to revise the manuscript in line with the reviewers' suggestions, and my previous amendments. We all find the manuscript much improved in scope and clarity, and would like to recommend publication following the amendments requested by Reviewer 2, and those listed below:

L11-14: an extremely long sentence featuring too many 'and's. Please divide the sentence for easier digestion.

L14: erroneous semicolon – should be a colon, or just a comma.

L16: 'meltwater in southwest sites' – should this be 'from' or 'at'?

L64: not sure this sentence is necessary: the majority of our readers know that there are no other ice sheets remaining in the northern hemisphere. If you choose to retain, either Northern should not be capitalised, or both Northern and Hemisphere should be capitalised (I think the latter?).

L91: typo

L119: spacing error prior to ref

P12: two paragraphs beginning with 'While', and multiple sentences beginning the same – suggest rewording one or two for readability.

L500: typo - missing space

Thank you for your contribution, I look forward to accepting the final version for publication shortly.

Best regards,

Liz Bagshaw